# Task Generalization with Stability Guarantees via Elastic Dynamical System Motion Policies

**Tianyu Li**
University of Pennsylvania
tianyuli@seas.upenn.edu

**Nadia Figueroa**
University of Pennsylvania
nadiafig@seas.upenn.edu

**Abstract:** Dynamical System (DS) based Learning from Demonstration (LfD) allows learning of reactive motion policies with stability and convergence guarantees from a few trajectories. Yet, current DS learning techniques lack the flexibility to generalize to new task instances as they overlook explicit task parameters that inherently change the underlying demonstrated trajectories. In this work, we propose Elastic-DS, a novel DS learning and generalization approach that embeds task parameters into the Gaussian Mixture Model (GMM) based Linear Parameter Varying (LPV) DS formulation. Central to our approach is the Elastic-GMM, a GMM constrained to SE(3) task-relevant frames. Given a new task instance/context, the Elastic-GMM is transformed with Laplacian Editing and used to re-estimate the LPV-DS policy. Elastic-DS is compositional in nature and can be used to construct flexible multi-step tasks. We showcase its strength on a myriad of simulated and real-robot experiments while preserving desirable control-theoretic guarantees. Project website: https://sites.google.com/view/elastic-ds.

**Keywords:** Stable Dynamical Systems, Reactive Motion Policies, Learning from Demonstrations, Task Parametrization, Task Generalization

## 1  Introduction

With advanced development in robotics and autonomous systems in the past decades, the opportunities and demands for more complex physical human-robot interaction (pHRI) in our everyday unconstrained environments are rising; thus, it is critical for robots to be adaptive, compliant, reactive, safe and easy to program [1, 2, 3]. In many cases, robots will need to acquire new skills to satisfy task requirements in an ever-changing environment. It is usually difficult for non-experts to program robots for complex motion tasks and even tedious for experts to reprogram them when task requirements change. A straightforward and intuitive approach for robots to develop new skills is through Learning from Demonstration (LfD) [4, 5, 6, 7, 8]. This paradigm allows robots to acquire skills, typically encoded or defined in literature as action policies, motion policies, or imitation policies, directly from motion examples provided by humans or even other robots, mirroring a teacher-student relationship.

In recent years, significant progress has been made in using LfD to learn complex and diverse motion tasks. However, many focused on learning and executing tasks from static or unchanged scenarios/environments/contexts, which could lead to failures when faced with out-of-distribution cases. From the machine learning perspective, this is the covariate shift issue that exists in many supervised learning related tasks, especially in the behavior cloning (BC) approach [7, 9]. By providing a fixed training dataset beforehand, the LfD algorithm will learn a policy that performs well for the training dataset but could fail to generalize to unseen input during deployment. The learned policy will become invalid due to the change of distribution. Hence, instead of memorizing human demonstrations for one scenario, the robot should be able to adapt and generalize to novel scenarios with satisfactory performance, given the same task objective.

7th Conference on Robot Learning (CoRL 2023), Atlanta, USA.

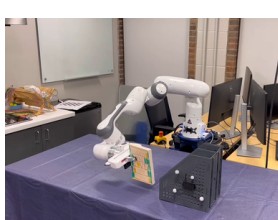 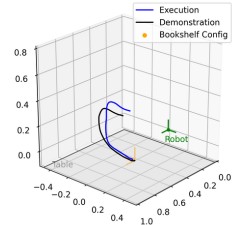 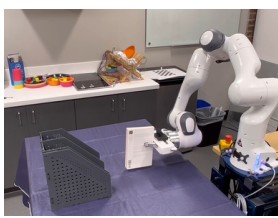 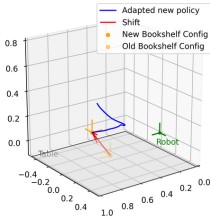

(a) Executing the learned DS  (b) Trajectory for the original learned policy  (c) Generalize the learned policy to new configuration  (d) Trajectory for the new policy

Figure 1: Task generalization for bookshelf stacking with Elastic-DS. (a-b) Given a single Demonstration, the Elastic-DS can efficiently learn and reproduce and (c-d) efficiently adapt to position and orientation changes in task parameters. Our Elastic-DS approach generalizes seamlessly and efficiently to new task parameter configurations while retaining stability and convergence guarantees.

**The Trilemma** - *Generalization* or adaptation abilities are particularly important to enable robots to perform effectively in dynamic environments. There are many attempts at generalization with methods like BC [10], Inverse RL (IRL) [11, 12, 13], Meta-Learning [14], Multi-Task Learning [15], Transfer Learning [16], Multi-Task Reinforcement Learning [17], Lifelong Learning [18, 19, 20], and continual learning [21]. However, the aforementioned approaches have no emphasis on providing *control-theoretical guarantees* on the learned policies, such as stability, boundedness, and convergence, all of which are critical for safe pHRI. On the other hand, the Dynamical System-based (DS) motion policy approach [3] offers many advantages such as reactivity, motion-level adaptation, and, most importantly, *stability guarantees*; and can be learned from only handful of demonstrations ensuring *minimal human effort* [22, 23]. However, due to the closed-form and offline learning nature of DS-based motion policies, they have no flexibility for generalizing to novel environments as they ignore explicit task parameters that inherently change the underlying trajectories that shape the DS vector fields. This limits their generalization capability and adoption as low-level policies. Hence, invoking the no free lunch theorem, we posit that the state-of-the-art currently suffers from the *generalization vs. stability vs. effort trilemma*.

**Goal** In this work, we seek to alleviate this trilemma by proposing an LfD approach that has i) stability guarantees, ii) the flexibility to generalize motions across novel scenarios, while iii) requiring minimal human effort during learning and adaptation/generalization, as depicted in Figure 1.

**Related Work** The number of techniques that exist for LfD/IL is vast [6, 4, 7, 8]. This work follows the BC approach, which learns a policy that maps states (state-action pairs, trajectories, and other contexts are also used) to control inputs [24]. DAgger [25] addressed distribution shift issues with online interactions/corrections. Whereas the generative adversarial learning framework [26] randomly explores for corrections that bring the policy close to the demonstrated distribution. These approaches offer generalization in terms of distribution shift but require either constant human effort or lots of data and computation and hold no control-theoretic guarantees on the learned policy. The recently introduced TaSIL (Taylor Series IL) framework introduces a simple augmentation to the BC loss such that the trained policy is robust to distribution deviations by ensuring incremental input-to-state stability, also benefiting from reduced sample-complexity [27]. Nevertheless, it cannot generalize to novel task instances/environments not seen during training. A Probably Approximately Correct (PAC)-Bayes IL framework was introduced in [10] that computes upper bounds on the expected cost of policies in novel environments. Prior works also focus on explicit skill generalization for novel environments or task instances, such as multitasking learning [28, 29, 30, 31] and meta-learning [14, 32]. While capable of generalizing learned tasks to different environments, these works require a considerable amount of offline/online training and require excessively large DNNs, which cannot be used in a reactive manner nor offer any form of control-theoretic guarantees.

A significant body of work tackles the generalization problem by emphasizing task parametrization (TP) as relevant task frames in SE(3) assigned to relevant objects in demonstrations, like TP-GMM [33, 34], TP-DMP [35], task invariants [36, 37] and environmental constraints [38]. Other

works focus on the motion policy parameter perspective, such as adapting explicit start and goal positions in movement primitives [39], conditioning on probability distributions like the Probabilistic Movement Primitives (ProMPs) for different via-points [40] and geometric descriptor [41]. While such works have demonstrated the ability for task generalization, few provide real-time reactive motion and stability guarantees, and most of them rely on the availability of demonstrations in different contexts or environments to extract the relevant task parameters for generalization.

**Approach** We propose a zero-shot approach for generalizing motion policies to novel scenarios while guaranteeing control-theoretic properties for safe deployment in pHRI. To achieve *stability*, we adopt the DS-based LfD paradigm [3] that learns motion policies as time-invariant nonlinear DS with Lyapunov stability guarantees. To achieve *generalization*, we follow the task-parametrization perspective [33, 34, 35] and propose to embed relevant task frames directly into the DS policy. While several neural network (NN) based formulations for stable DS motion policies exist, such as neurally imprinted vector fields [42], DNN via contrastive learning [43], diffusion models [44], and euclideanizing flows [45]; given their black-box nature it is not straightforward to embed such task parameters into these formulations. Further, NN approaches need multiple demonstrations to encode the stable DS properly. To achieve *minimal data, compute and human effort* during learning, we adopt the Gaussian Mixture Model (GMM) based Linear Parameter Varying (LPV-DS) formulation [22, 3] which has shown to be computationally efficient and capable of learning stable vector fields of complex motions from a single demonstration [23]. Finally, we take inspiration from elastic bands [46] and trajectory editing [47] and propose a novel approach to generalize the GMM-based LPV-DS to novel scenarios without new demonstrations, referred to as Elastic-DS.

**Contributions** We introduce the Elastic-DS formulation as a solution to the LfD trilemma (See Figure 2). The Elastic-DS is constrained to a set of task parameters described as geometric descriptors representing the invariant features of a task (e.g., object, via-point, or target configurations). It is capable of efficiently generating novel DS policies upon task parameter changes without requiring new data or human input for single, multi-step tasks and the composition of new tasks via DS stitching.

## 2 Problem Statement

Let $\mathcal{D} := \{\{\xi_{t,n}, \dot{\xi}_{t,n}\}_{t=1}^{T_n}\}_{n=1}^{N}$ be a set of $N$ demonstration trajectories collected from kinesthetic teaching for a task, where $\xi_t \in \mathbb{R}^d$, $\dot{\xi}_{t,n} \in \mathbb{R}^d$ represent the kinematic robot state and velocity vectors at time $t$, respectively, for the $n$-th trajectory with length $T_n$. In this work, we consider $\xi_t \in \mathbb{R}^d$ to be the end-effector Cartesian position. Let $\dot{\xi} = f(\xi)$ be a first-order DS that describes a motion policy in $\mathbb{R}^n$ state space. Given $\mathcal{D}$, the goal is to infer $f(\xi) : \mathbb{R}^n \to \mathbb{R}^n$ such that any point $\xi$ in the state space leads to a stable attractor $\xi^*$, with $f(\xi)$ described by a set of parameters $\Theta$ and $\xi^*$

$$\dot{\xi} = f(\xi; \Theta, \xi^*) \Rightarrow \lim_{t \to \infty} \|\xi - \xi^*\| = 0. \tag{1}$$

Usually, such DS motion policies are learned in an offline manner and fixed for the execution phase [23]. The main contribution of this work is to introduce an update stage that parametrizes the system further, allowing the motion policies to adapt and generalize to new tasks, as shown in Figure 2.

**Task Parameters** A task is defined as a combination of multiple trajectories where each of them is grounded on a geometric constraint descriptor set $O_i = \{o_{enter}, o_{exit}\} \in SE(3)^2$, describing the two endpoint poses. Let $\beta_i \in \mathbb{R}^{d \times (k+1)}$ be generalization parameters in the state space conditioned on the geometric descriptors $O_i$. As $O_i$ changes, $\beta_i$ will change accordingly to generate a new set of DS to reach $O_i$ with the correct poses. In this work, the geometric descriptors $O_i$ is assumed to be known or given during demonstrations. However, it could come from various upstream sources such as human specifications [23] or generative segmentation algorithms [48].

**Motion Policy** We propose the following motion policy for task-parameterized generalization:

$$\dot{\xi} = \sum_{i=1}^{M} \delta(\xi, O_i) f_i(\xi; \Theta_i, \beta_i, \xi_i^*) \tag{2}$$

where $\delta(\xi, O_i)$ is an activation function determining the sequence of execution for $M$ DSs describing a multi-step sequential tasks. Hence, given $\mathcal{D}$ with the same behavior and $M$ geometric descriptor

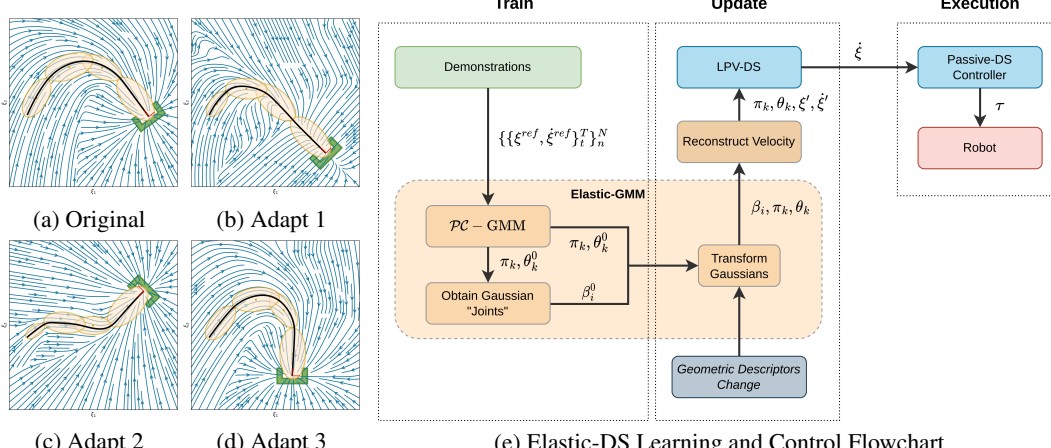

Figure 2: (a) Elastic-DS reproducing a stable LPV-DS vector field on the originally demonstrated data. (b-d) Elastic-DS generating stable LPV-DS motion policies from peg attractor changes (translation and rotation) without new demonstration. (e) Flowchart of Elastic-DS approach.

$O_i$ configurations, our approach finds $\beta_i$ that will generate new DS motion policies with i) stability guarantees with respect to their corresponding attractors $\xi_i^*$, and ii) the flexibility to achieve the same task with new geometric constraint descriptor configurations.

## 3 Preliminaries: $\mathcal{PC}$-GMM and LPV-DS Motion Policy [22]

The GMM-based LPV-DS [22] motion policy has the following formulation,

$$\dot{\xi} = f(\xi) = \sum_{k=1}^{K} \gamma_k(\xi)\left(A^k\xi + b^k\right) \quad \text{s.t.} \begin{cases} \left(A^k\right)^T P + PA^k = Q^k, Q^k = \left(Q^k\right)^T \prec 0 \\ b^k = -A^k\xi^* \end{cases} \tag{3}$$

where $\gamma_k(\xi)$ is the state-dependent mixing function that quantifies the weight of each linear time-invariant (LTI) system $(A^k\xi + b^k)$. $\mathcal{N}(\xi|\theta_k)$ denotes the probability of observation $\xi$ from the $k$-th Gaussian component parametrized by $\theta_k = \{\mu_k, \Sigma_k\}$, and $\pi_k$ represents the prior probability of an observation from this particular component and the a posteriori probability is

$$\gamma_k(\xi) = \frac{\pi_k\mathcal{N}(\xi|\theta_k)}{\sum_{j=1}^{K}\pi_j\mathcal{N}(\xi|\theta_j)} \quad \text{from} \quad p\left(\xi|\{\pi_k, \theta_k\}\right) = \sum_{k=1}^{K}\pi_k\mathcal{N}(\xi|\mu_k, \Sigma_k). \tag{4}$$

Intuitively, Eq. 3 fits a mixture of linear DS to a complex non-linear trajectory, with $\gamma_k(\xi)$ ensuring the smoothness of the reproduced trajectories. Hence, each Gaussian component must be placed on quasi-linear segments of $\mathcal{D}$. With $\mathcal{PC}$-GMM, the optimal number of linear DS $A^k$ for a given $\mathcal{D}$ can be automatically inferred.

**Stability** To guarantee global asymptotic stability of Eq. 3, a Lyapunov function $V(\xi) = (\xi - \xi^*)^T P(\xi - \xi^*)$ with $P = P^T \succ 0$, is used to derive the stability constraints in Eq. 3. Minimizing the fitting error of Eq. 3 with respect to demonstrations subject to constraints in Eq. 3 yields a non-linear DS with a stability guarantee via a Semi-Definite Program (SDP) [22]. Implementation details are provided in Appendix A.

## 4 Elastic-DS

### 4.1 Elastic-GMM

To adapt generalization parameters $\beta_i$ for the corresponding spatial change in the geometric descriptors $\mathcal{O}_i$, we introduce Elastic-GMM as the core component of augmenting LPV-DS (Eq. 3) into Elastic-DS (Eq. 2). Figure 2e shows the pipeline of Elastic-DS. During the training stage, we use $\mathcal{PC}$-GMM [22] to obtain a set of initial Gaussian parameters $\{\pi_k, \theta_k^0\}$ as well as the initial $\beta_i^0$. The update stage will produce the updated Gaussian parameters $\{\pi_k, \theta_k\}$ as well as the updated $\beta_i$, which are key way-points in the state space. A trajectory will be generated based on the key points to specify the velocity for the DS after the update. With the new Gaussian parameters and

the new velocity information, an updated LPV-DS can be learned. If further updates happen in the environment, we can directly update LPV-DS using the transform without re-estimating the GMM, which avoids a time-consuming stage. During the execution, a passive-DS controller [3] will take the newest LPV-DS output velocity $\dot{\xi}$ to generate the corresponding joint torque $\tau$ for the robots. The upcoming section will discuss each key component in detail. The algorithm is in Appendix D.

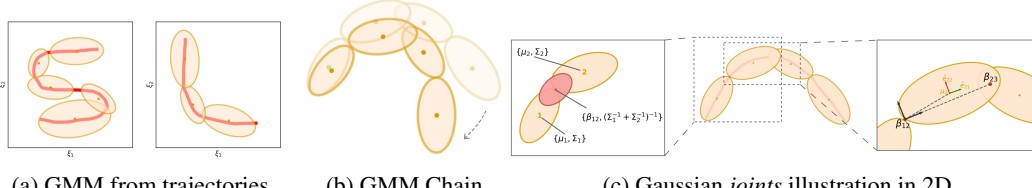

(a) GMM from trajectories      (b) GMM Chain      (c) Gaussian *joints* illustration in 2D

Figure 3: Illustrations of $\mathcal{PC}$-GMM fitted on trajectory data (a) and the Gaussian chain transform from Section 4.1.1, (c) a closer look at the Gaussian *joints* introduced in Section 4.1.3.

### 4.1.1 GMM Chain

Following the LPV-DS pipeline, the demonstration trajectory is encoded into GMM $\{\pi_k, \theta_k^0\}$ using $\mathcal{PC}$-GMM [22]. As shown in Figure 3a, a trajectory is extracted and simplified into a *chain of Gaussians links*. In the update stage (Figure 2e), we transform the spatial relationships among the Gaussians to achieve task adaptation. Imagine an analogy of the Gaussian chain being a robot arm, which could be rotated around each joint to achieve a specific geometric configuration as shown in Figure 3b. Note the robot arm analogy is on the end-effector trajectory instead of the actual robot arm. The generalization parameters $\beta_i$ are the *joints* between each pair of the neighboring Gaussians, which describes the spatial relationship between the neighbors. After the $\mathcal{PC}$-GMM step, we can obtain the initial $\beta_i^0$ and later update the $\beta_i^0$ to $\beta_i$ to achieve transform. The *joint* between two Gaussians, which is the $\beta_i$, is the mean of the product between them as described by the picture on the left in Figure 3c, which could be obtained by,

$$\Sigma_t = (\Sigma_1^{-1} + \Sigma_2^{-1})^{-1} \qquad \beta_{i,12} = \Sigma_t(\Sigma_1^{-1}\mu_1 + \Sigma_2^{-1}\mu_2) \tag{5}$$

where $\mu$ and $\Sigma$ are the mean and covariance of the Gaussians $\{\pi_1, \theta_1^0\}$ and $\{\pi_2, \theta_2^0\}$. To complete the robot arm analogy, we also need to determine the *links* position and orientation $T_{GMM} \in SE(3)^K$ with respect to the *joints*. Figure 3c depicts the Gaussian mean position $\mu_j$ and orientation (described by the eigenvectors $\hat{e}_j$ of the covariance matrix $\Sigma_j$) with respect to the frame at the last joint with the x-axis pointing towards the next joint (in the direction of the demonstration). All of the above are constructed as the initial condition, in which no update is involved yet. Later when the states of the generalization parameter $\beta_i$ change, we will recover the same transformation of the mean and covariance with respect to the corresponding $\beta_i$. Before introducing the approach to obtain the new *joint* positions $\beta_i$ we provide a brief summary of the Laplacian editing approach [49, 47].

### 4.1.2 Laplacian Editing Primer

Laplacian Editing allows directly modifying an existing trajectory defined by $m$ waypoints $\boldsymbol{r} \in \mathbb{R}^{d \times m}$ while capturing local properties. First, we convert the waypoints $\boldsymbol{r}$ in cartesian space into Laplacian coordinates $\Delta$ with the graph Laplacian matrix $L \in \mathbb{R}^{m \times m}$ [47],

$$L_{ij} = \begin{cases} 1 & \text{if } i = j, \\ -\dfrac{w_{ij}}{\sum_{j \in \mathbf{N}_i} w_{ij}} & \text{if } j \in \mathbf{N}_i, \\ 0 & \text{otherwise.} \end{cases} \tag{6}$$

where $\mathbf{N}_i$ are a set of neighbor points $r_j$ for waypoint $r_i$, and $w_{ij}$ is a weight set to 1 for this work. One can obtain $\Delta = L\boldsymbol{r}$, where $\Delta$ is a concatenation of the Laplacian coordinate for each waypoint $\delta_i = \sum_{j \in \mathbf{N}_i} \frac{w_{ij}}{\sum_{j \in \mathbf{N}_i} w_{ij}} (r_i - r_j)$. The matrix $L$ can be singular, so one can impose constraints on the system $L\boldsymbol{r} = \Delta$ when solving for new waypoints $\boldsymbol{r}$ to achieve editing [47].

### 4.1.3 Transform Gaussians with Constraints

The initial joints $\beta_i^0$ are converted into the Laplacian coordinate as in Section 4.1.2 and used to construct a least-square objective in Eq. 7. Then, we align the first link (formed by $\beta_0$ and $\beta_1$) and the last link (formed by $\beta_{n-1}$ and $\beta_n$) with the geometric descriptor $\mathcal{O}_i$, which forms the constraints for the least-square formulation. When solving for this optimization, the other $\beta_i$ will adjust based on the Laplacian objective, softly preserving local position properties,

$$\min_{\beta_i} J\left(\beta_i\right) = \|L\beta_i - \Delta\|_2^2 \quad \text{subject to} \quad \begin{cases} T_{0,1}(\beta_{i,0}, \beta_{i,1}) = O_{start} \\ T_{n-1,n}(\beta_{i,n-1}, \beta_{i,n}) = O_{end} \end{cases} \tag{7}$$

where $L \in \mathbb{R}^{n \times n}$ and $\Delta$ are the same as in Section 4.1.2. $T_{0,1}(\beta_0, \beta_1)$ represents the frame transformation from $\beta_{i,0}$ to $\beta_{i,1}$. The solution of this optimization will produce new *joints* positions $\beta_i$. After that, the *link* position and orientation (which are the Gaussians' means and covariances), as well as the scale, are recovered using the $T_{GMM}$ recorded from the previous section. The orientation of the Gaussian is determined by the eigenvector shown in red and green in Figure 3c. Each orientation will remain fixed with respect to the last joint frame ($\beta_{12}$ in Figure 3c). If a rotation happens to the last joint frame, the Gaussian frame associated with the last joint will be moved together in the global frame but remain the same in the last joint frame. The scale is determined by the change from the original distance between each pair of neighboring *joints*, which scales the eigenvalues of the Gaussians' covariances. Referring back to the flowchart in Figure 2e, this section outputs the updated *joint* positions $\beta_i$ and updated GMM parameters $\theta_k$ ($\pi_k$ stay unchanged).

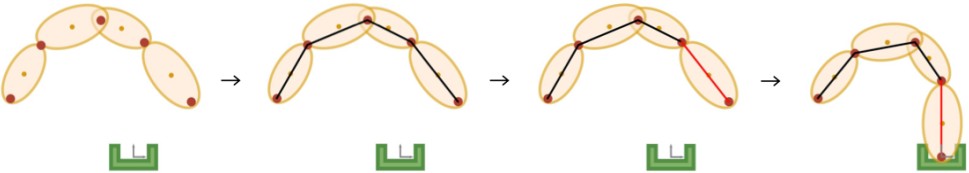

Figure 4: After obtaining the joints between neighboring Gaussians, a piecewise linear trajectory is formed. The green polygon is the geometric descriptor at the exit of the trajectory. We set a constraint on the last segment to align with the pose of the geometric descriptor. After solving (7), we achieve the final transformation (depicted in the rightmost image). Note: This illustration only shows the constraint at the exit, while in general, the constraints could be at the entry and the exit.

### 4.2 Create Velocity Profile

Depending on the new task constraints, the velocity requirement could be different from the original demonstration. Therefore, this approach offers the opportunity to modify the velocity by regenerating a trajectory along the Gaussians *joints* as the waypoints. There are many ways to achieve this with known waypoints, such as splines or minimum jerk trajectory. We provide a simple example of using Laplacian Editing to generate a new trajectory. First, we connect $\beta_{i,0}$ and $\beta_{i,n}$ to form a linear trajectory $\zeta \in \mathbb{R}^{d \times p}$. $p$ is the number of points on this trajectory. We then force this trajectory to pass through $\beta$, the Gaussian joints, with Laplacian editing,

$$\min_{\zeta} J\left(\zeta\right) = \|L_\zeta \beta - \Delta_\zeta\|_2^2 \quad \text{subject to} \quad \zeta_j = \beta_{i,q} \tag{8}$$

where $j \in \{0, ..., p-1\}$ is the index in $\zeta$ for matching the corresponding $\beta_i$. More details can be found in Appendix B. The velocity will be determined by the finite difference between the edited trajectory neighboring data points divided by the $dt$ collected from the demonstrations. After this section, the velocity information and the updated GMM will become the input for learning a new DS motion policy following Section 3, which by nature preserves the stability guarantees.

### 4.3 Multiple Segments

Multiple Elastic-DS could be stitched together to achieve a via-point trajectory and even long-horizon multi-segment tasks. The index $i$ in the task parameters $\beta_i$ represents multiple segments,

consequently multiple DSs in (2) when $M > 1$. To allow adding spatial constraints along the trajectory, one can split the task into multiple segments and process them in a divide-and-conquer manner. There are two possible task-specific cases for stitching the segments: (i) The activation function $\delta(\xi, O_i)$ is in charge of the switch for multiple DSs. The next DS will be activated by $\delta$ when the last DS reaches the attractor. (ii) To create a smooth movement, this case will first connect all the Elastic-GMMs from different segments and then learn a single DS. The $\delta$ function is not in use for this case. As mentioned in Section 2, the interesting separation points described by the geometric descriptors are specified by some upstream sources. For more information about the split and stitching process, please refer to Appendix C. The flexibility of composing and regrouping different transformed DS with the new constraint poses allows the possibility for multi-task scenarios.

## 5 Experimental Results

### 5.1 2D Experiments

This section shows 2D examples of using Elastic-DS. The learned DS is plotted as steamplots describing a velocity vector field (in blue) with GMM (in orange) geometric descriptors (in green) and rollout trajectory (in black) overlaying on top. The 2D simulation in Figure 5 shows the atomic case of one segment trajectory conditioned on a geometric descriptor with constraints at the endpoints. The two ends of the geometric descriptor (green polygons) are shifted and rotated to show different configurations and the changes in the DS vector field. Figure 6 shows the example of using a via-point to modify the policy in the middle of a single DS corresponding to case (ii) in section 4.3. The DS motion policy is able to adapt to the changes. For more details about stitching the trajectory, please refer to Appendix C. Figure 7 and Table 1 display a comparison to TP-GMM-DS [50, 51], TP-GPR-DS [52], and TP-proMP [52, 40]. Appendix E shows the failure cases of TP-GMM with fewer demonstrations. With a single demonstration, Elastic-DS is able to generalize well, while other methods fail. For more comparison details, please refer to Appendix F.

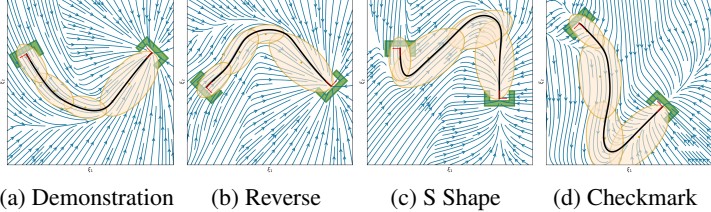

(a) Demonstration    (b) Reverse    (c) S Shape    (d) Checkmark

Figure 5: Elastic-DS motion policy generalizes based on changes of start and goal poses

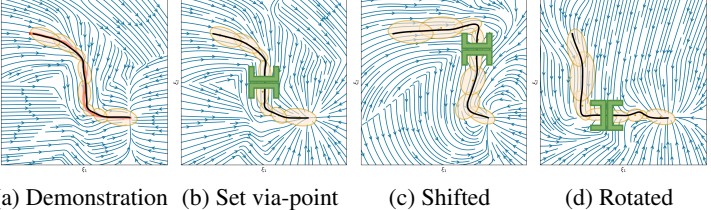

(a) Demonstration    (b) Set via-point    (c) Shifted    (d) Rotated

Figure 6: Elastic-DS motion policy generalizes based on changes of the via-point

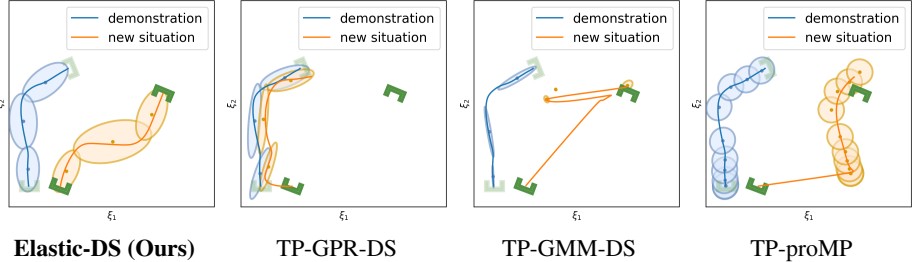

**Elastic-DS (Ours)**    TP-GPR-DS    TP-GMM-DS    TP-proMP

Figure 7: Elastic-DS is able to handle the new situation while the other methods fail with a single demonstration. Appendix E shows that TP approaches require more demonstrations to succeed.

| Metric | Elastic-DS (Ours) | TP-GPR-DS | TP-GMM-DS | TP-proMP |
|---|---|---|---|---|
| Start Cosine Similarity ↑ | **0.9843** | -0.3981 | 0.9405 | 0.4777 |
| Goal Cosine Similarity ↑ | **0.9998** | 0.5453 | 0.6324 | 0.8965 |
| Endpoints Distance ↓ | **0.0008** | 0.8360 | 0.07642 | 0.1788 |

Table 1: The metrics include the orientation alignment of the start and goal as well as the position distance with the geometric descriptors in the new instance. The resulting data corresponds to the orange trajectories in Figure 7. With both ends moving, Elastic-DS remains in good performance on all three metrics. More details about the metrics are in Appendix F.

## 5.2 Robot Experiments

We demonstrate the validation result through four different real robot experiments on the Franka Emika Panda robot: Bookshelf, Pick and Place, Tunnel, and Combination, see Figure 2, 8 and Appendix G. In these experiments, the geometric descriptors are detected from a motion capture system. By attaching motion capture markers on objects, we specify geometric descriptors anchored on the objects of interest. Three experiments will start with a human performing kinesthetic teaching by moving the end-effector. Then, the execution of the original learned DS from the demonstration will be shown. The objects of interest will then be shifted and rotated with different configurations. The last experiment shows the ability to compose new tasks. Without any new demonstration, the robot can still achieve the required tasks. Please refer to the video and Appendix G.

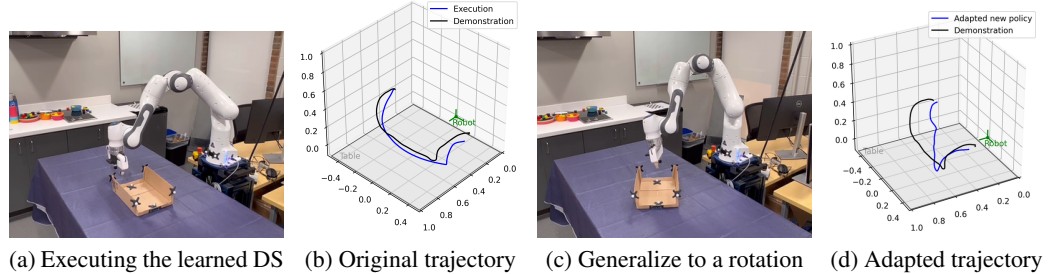

(a) Executing the learned DS (b) Original trajectory (c) Generalize to a rotation (d) Adapted trajectory

Figure 8: Example of Elastic-DS adaptation in a tunnel passing task

## 6 Conclusions and Limitations

We propose Elastic-DS in this work, which allows modifying DS with task parameters conditioned on geometric features to achieve task generalization. As the core component, we introduced Elastic-GMM to augment the original LPV-DS to create more flexible task-specific motions with as low as one demonstration. By showing both 2D simulation and different 3D robot experiments, we validate the ability of Elastic-DS to perform task generalization as well as the potential for multi-task and long-horizon motion policies. Following we discussed the limitations of our approach.

**Limitations** First, this work only considers end-effector motions in the Cartesian position space. The task constraints are only for translational motion as well. To achieve more variety of tasks and extend to more possible poses, the orientation space has to be considered. Further directions could adopt works like [53] and [54] to produce DS motion policies and meet task constraints in the full pose space. To go even further, the full pose could also contain the gripper state, which could be achieved with a coupled DS approach such as [55] and [3]. Motion policy in the joint space should also be considered [56]. Second, the geometric descriptors are assumed to be given by human specification in this work. To address this limitation, we could utilize rigid object tracking methods like BundleTrack [57] to identify the geometric interactions between the robot demonstration and objects. Finally, the task adaptation in this work is fast yet not real-time ($\approx 1s$ for 2D and 3D data on a typical laptop with Intel i7-12700H and 16GB memory, depending on the complexity of the task). Computation time analysis is provided in Appendix H. With the dynamic nature of physical human-robot interaction, it is important to provide continuous adaptation on the fly to create a seamless experience. Hence, our immediate next step is to accelerate adaptation time to ms scale.

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

# Appendix

## A  LPV-DS Parameter Optimization

**GMM-based LPV-DS formulation** It is described in Section 3 Preliminaries: $\mathcal{PC}$-GMM and LPV-DS Motion Policy.

$$\dot{\xi} = f(\xi) = \sum_{k=1}^{K} \gamma_k(\xi)\left(A^k\xi + b^k\right) \quad \text{s.t.} \ \begin{cases} \left(A^k\right)^T P + PA^k = Q^k, Q^k = \left(Q^k\right)^T \prec 0 \\ b^k = -A^k\xi^* \end{cases} \tag{9}$$

**DS Estimation** The set of DS parameters $\theta_{DS} = \{A^k, b^k\}$ for $f(\xi)$ is estimated with LPV-DS by minimizing the Mean Square Error (MSE) against the demonstrations [22] subject to stability constraints in Equation 9.

$$\min_{\theta_{DS}} J\left(\theta_{DS}\right) = \sum_{n=1}^{N_{\text{ref}}} \sum_{t=1}^{T_N} \left\| \dot{\xi}_{t,n}^{\text{ref}} - f\left(\xi_{t,n}^{\text{ref}}\right) \right\|^2 \tag{10}$$

$$\text{s. t.} \ \begin{cases} \left(A^k\right)^T P + PA^k = Q^k, Q^k = \left(Q^k\right)^T \prec 0 \\ b^k = -A^k\xi^* \qquad \forall k = 1, \dots, K \end{cases}$$

which is a constrained non-convex semi-definite program (SDP). Further, when $P$ is known (or estimated beforehand as in [22]) the problem becomes a convex SDP that can be solved highly efficiently with off-the-shelf QP solvers [22, 3].

# B Creating Velocity Profile with Laplacian Editing

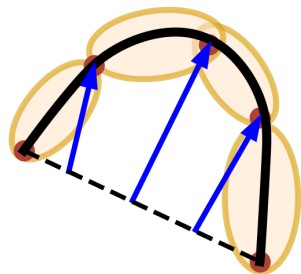

Figure 9: Connect endpoints and set constraints along the trajectory progress at the Gaussian joints

The main goal is to force a linear trajectory $\zeta \in \mathbb{R}^{d \times p}$ to pass through $\beta$, the Gaussian joints, with Laplacian editing,

$$\min_{\zeta} J(\zeta) = \|L_\zeta \beta - \Delta_\zeta\|_2^2 \quad \text{subject to} \quad \zeta_j = \beta_{i,q} \tag{11}$$

We first calculate the total Euclidean distance along the piecewise connected line between neighboring *joints* with total distance $D = \sum_{i=0}^{n-1} \|\beta_{i+1} - \beta_i\|$. Then we have the percentage $\lambda_q$ of the progress that the *joints* positions in $\beta_i$ make along the total distance $D$. By using $\lambda_q$, the corresponding $\beta_i$ is mapped to the index $j = \lfloor \lambda_q(p-1) \rfloor$. In practice, by also enforcing constraints between the *joints* with linearly interpolation will help the trajectory become more aligned with the GMM link and the geometric descriptors. The velocity will be determined by the finite difference between the edited trajectory neighboring data points divided by the $dt$ collected from the demonstrations. One can specify the velocity by controlling the spacing between the edited trajectory neighboring data points and the number of points $p$.

# C Stitching from Multiple Segments

As mentioned in the main paper, there are two design trade-off approaches for stitching multiple segments. Depending on the task, one can choose either to learn a DS for each Elastic-GMM (Sequential DS) or learn a single DS for the stitched Elastic-GMM (Combined DS). Figure 10a shows the flow describing the former case, and Figure 10b shows the flow for the latter case.

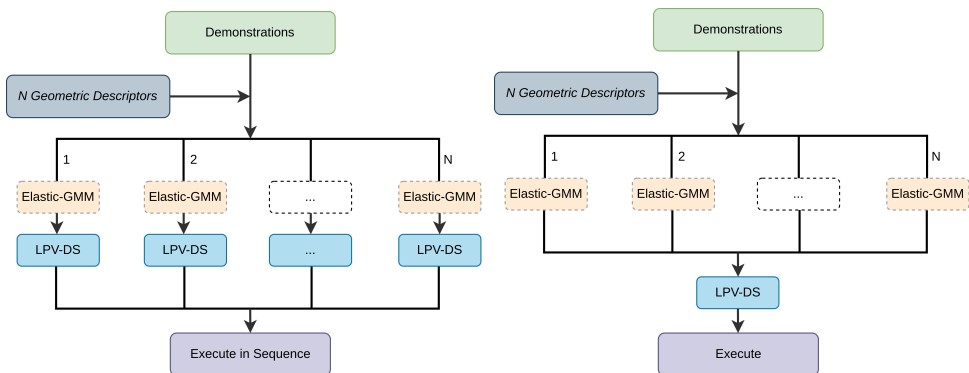

(a) This flow ensures that the position geometric constraints will be reached for the task with multiple DS

(b) This flow produces smooth motion with a single DS. One can adapt [23] to achieve task satisfaction.

Figure 10: Different task split flows for design trade-off.

Here is an example of the two cases in which they are used to modify a DS with via-point. By specifying an interesting point in the demonstration (usually by human specification or upstream computer vision method), the trajectory will be split into separate components. Each individual segment will be processed with Elastic-GMM to meet the new interesting via-point geometric constraints as depicted in green polygons. By performing such steps, we can pose constraints not just on the endpoints of the demonstration but also on the intermediate points of the demonstration. This via-point experiment shows that with changes in the via-point constraints, Elastic-DS can adapt to them.

## C.1 Sequential Elastic-DS

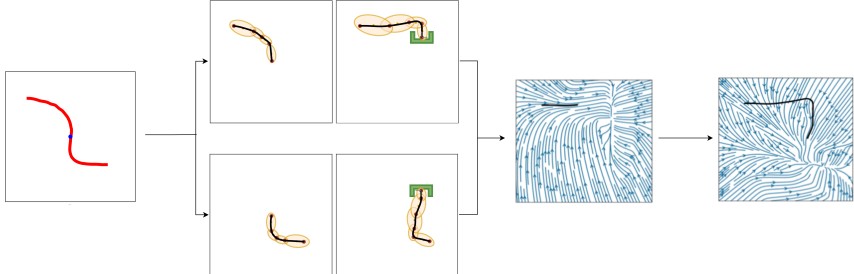

Figure 11: An example of the pipeline for sequential DS. In this case, the single demonstration (in red) is separated at the blue spot. Both segments perform Elastic GMM to meet the geometric descriptor constraints and learn DS individually. The end results are two DS which will run sequentially. After the first DS reaches its attractor, the robot will start to follow the second DS. This corresponds to the flowchart in Figure 10a

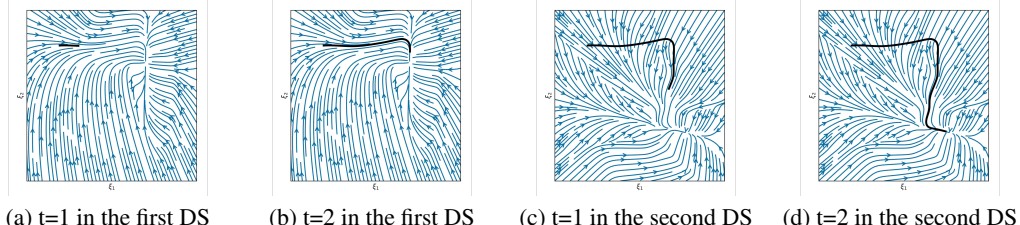

(a) t=1 in the first DS    (b) t=2 in the first DS    (c) t=1 in the second DS    (d) t=2 in the second DS

Figure 12: Modified DS based on the via point as multi-segment DS without new demonstration. The plots here show the rollout trajectory of the switch DS. It uses the same GMM as in Figure 14c.

## C.2  Combined Elastic-DS

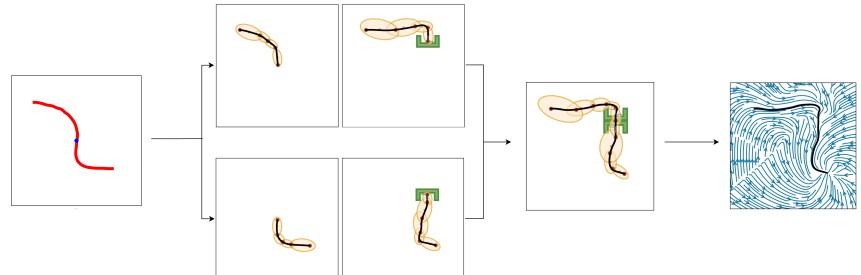

Figure 13: An example of the pipeline for building a single DS. In this case, the single demonstration (in red) is separated at the blue spot. After obtaining the GMM for each segment, they are stitched together and generate a single DS; such an approach will generate smooth velocity but could result in trajectories missing the geometric constraints with perturbation. One could adapt LTL-DS [23] to alleviate this and achieve task satisfaction. This corresponds to the flowchart in Figure 10b
.

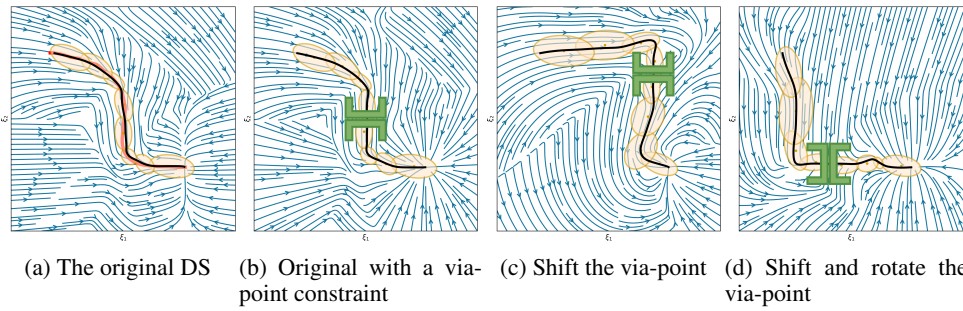

(a) The original DS    (b) Original with a via-    (c) Shift the via-point    (d) Shift and rotate the
                       point constraint                                    via-point

Figure 14: Modified DS based on the via point as a single DS without new demonstrations

# D    Transforming Elastic-GMM

---

**Algorithm 1:** Transform Elastic-GMM for Generalization

---

**Input:** $\{\mu_k, \Sigma_k\}_{k=1}^K, \xi_{t=1}, \xi_{t=T}, O_{start}, O_{end}$
**Output:** $\{\mu_k^{'}, \Sigma_k^{'}\}_{k=1}^K$
$n \leftarrow 2$;
$k \leftarrow 1$;
**while** $k \leq K - 1$ **do**
    $\Sigma_n = (\Sigma_1^{-1} + \Sigma_2^{-1})^{-1}$; {Eq (5) in the main text}
    $\beta_n = \Sigma_n(\Sigma_1^{-1}\mu_1 + \Sigma_2^{-1}\mu_2)$; {Eq (5) in the main text}
    $\lambda_i, \hat{e}_i = eig(\Sigma_k)$;
    $M_{n-1}$ = create a frame at $\beta_{n-1}$ using $\beta_n - \beta_{n-1}$ as the x-axis;
    $\zeta$ = create a frame with $\hat{e}_i, \mu_k$;
    $\Gamma_{k,n-1}$ = the transformation from $M_{n-1}$ to $\zeta$;
    $n = n + 1$;
    $k = k + 1$;
**end**
$\beta_1 = \xi_1, \beta_N = \xi_T$;
Construct L via Eq (6) in the main text;
$\Delta = L\beta$;
$T_{0,1}$ = Create a frame with $\beta_0$ and $\beta_1$ at $\beta_0$;
$T_{n-1,n}$ = Create a frame with $\beta_{n-1}$ and $\beta_n$ at $\beta_n$;
$\beta^{'} = \arg\min_\beta J(\beta) = \|L\beta - \Delta\|_2^2$    subject to constraints in Eq (7) in the main text;
$n \leftarrow 2$;
$k \leftarrow 1$;
**while** $k \leq K$ **do**
    Recover $\mu_k^{'}, \hat{e_{ki}}^{'}$ with $\Gamma_{k,n-1}$ w.r.t $\beta_{n-1}^{'}$;
    Scale $\lambda_{ki}^{'}, \mu_{kx}^{'}$ according to the new distance between neighboring $\beta$;
    Reconstruct $\Sigma_k^{'}$ with $\lambda_{ki}^{'}, \hat{e_{ki}}^{'}$;
    $n = n + 1$;
    $k = k + 1$;
**end**
Return $\{\mu_k^{'}, \Sigma_k^{'}\}_{k=1}^K$;

---

# E Failure Cases for TP-GMM Task-Parameterized Learning

This section shows the performance of task-parametrized policy learning under a different number of demonstrations. Specifically, the method shown here uses Task-Parameterized Gaussian Mixture Model as the encoding strategy and Gaussian mixture regression as the trajectory reproduction [52, 51]. There are two examples in total. Each example will start with four demonstrations (samples) in blue (but faded), moving from the bottom geometric descriptor (frame) to four various other geometric descriptors on top. For the new situation, the two geometric descriptors will be placed at new positions (in deep green). The orange trajectory shows the reproduction in the new situation, with the orange ellipses being the GMM encoding. The number of demonstrations will decrease in each example to show the generalization ability to the new situation with less training data as well as the sensitivity to the coverage of the training data.

## E.1 Case Example 1

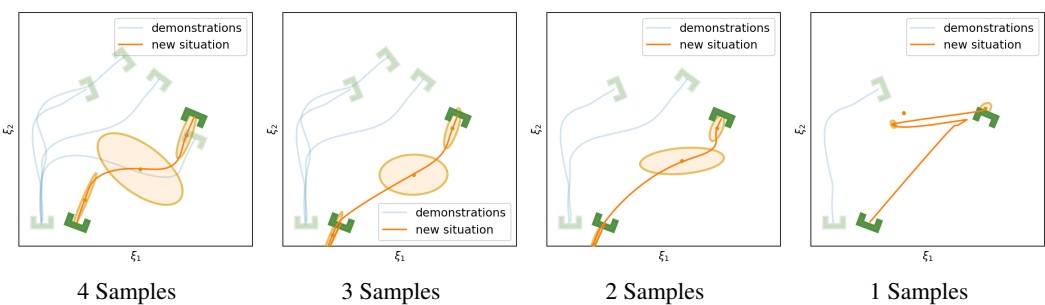

| 4 Samples | 3 Samples | 2 Samples | 1 Samples |

Figure 15: With four demonstrations and the new frames being placed among the samples, the new motion policy performs well. As the number of demonstrations decreases to three and two, the new motion policies still reach the goal with reasonable behaviors though the initial movements are in the opposite direction. It clearly does not perform well with a single demonstration.

## E.2 Case Example 2

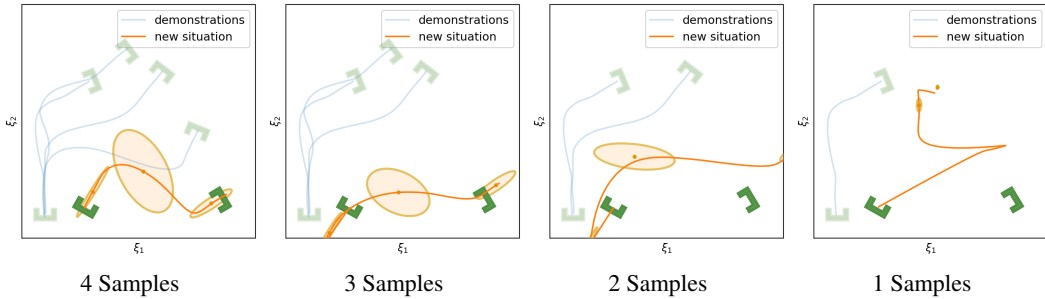

| 4 Samples | 3 Samples | 2 Samples | 1 Samples |

Figure 16: The new frames are placed further away from the demonstration's coverage area, but TP-GMM can still generate a correct reproduction with four demonstrations. However, as the number of demonstrations decreases, its performance decays and eventually fails with the single demonstration case.

In conclusion, TP-GMM does not generalize well when reducing the number of demonstrations in these two examples. It requires more effort to determine the appropriate placement of the demonstrations to generalize well. With a single demonstration, it tends to overfit that trajectory. The next section will show the performance of Elastic-DS being able to generalize well with a single demonstration compared to other TP approaches.

# F    Compare to Existing Methods

To show the advancement of our method, we create both qualitative and quantitative comparisons against various benchmark methods in the task-parametrized (TP) approach [52, 50]. Specifically, the benchmarks include Task-Parameterized Gaussian Process Regression with DS-GMR for motion reproduction (TP-GPR-DS) [52], Task-Parameterized Gaussian Mixture Model with DS-GMR for motion reproduction (TP-GMM-DS) [51, 50, 52], Task-Parameterized Probabilistic Movement Primitives (TP-proMP) [52, 40]. We use different quantitative metrics to show satisfaction in generalizing tasks:

- **Start Cosine Similarity**: It describes the starting direction of the trajectory and how it aligns with the entry/starting geometric descriptor. We take the first two data points to create a vector $v_s$ and compare it against the pointing direction of the entry/starting geometric descriptor $v_{Os}$. The closer this value is to one, the better.

$$\cos(\theta_s) = \frac{v_s \cdot v_{Os}}{\|v_s\|\|v_{Os}\|} \quad (12)$$

- **Goal Cosine Similarity**: It describes the goal reaching direction of the trajectory and how it aligns with the goal/exit geometric descriptor. We take the last two data points of the trajectory to create a vector $v_g$ and compare it against the pointing direction of the goal/exit geometric descriptor $v_{Og}$. The closer this value is to one, the better.

$$\cos(\theta_g) = \frac{v_g \cdot v_{Og}}{\|v_g\|\|v_{Og}\|} \quad (13)$$

- **Endpoints Distance**: Besides the pointing direction, it is important that the trajectory starts from the center of the starting geometric descriptor $P_{Os}$ and reach the center of the goal geometric descriptor $P_{Og}$. Let $\xi_0$ be the start of the trajectory and $\xi_T$ be the end of the trajectory. The metric will be the sum of the two Euclidean distances. The smaller this value, the better.

$$D = d(\xi_0, P_{Os}) + d(\xi_T, P_{Og}) \quad (14)$$

We compare four different trials with the same training data (a single demonstration): *Close*, *Far*, *Both Ends Shifted*, and *Both Ends Shifted Far* with increasing difficulty levels. Each subsection below describes a trial with a plot showing with red arrows how the geometric descriptors (in green) are being changed. Then it will be followed by four plots showing our method compared to three other methods. Each subsection will include a table showing the quantitative comparison. The single demonstration data is taken from the attached library code in [52]. The parameters for the benchmark methods remain in default as in the code from [52]. There are no required tuning parameters for Elastic-DS.

## F.1    Close

| Metric | Elastic-DS (Ours) | TP-GPR-DS | TP-GMM-DS | TP-proMP |
|---|---|---|---|---|
| Start Cosine Similarity ↑ | **0.9857** | -0.8222 | 0.9794 | 0.2062 |
| Goal Cosine Similarity ↑ | **0.9999** | 0.7397 | 0.5422 | 0.9889 |
| Endpoints Distance ↓ | **0.0008** | 0.4298 | 0.7564 | 0.3062 |

Table 2: Elastic-DS outperforms other methods for meeting the new geometric descriptor constraints. Its cosine similarities are the closest to 1, and the endpoints distance is the closest to 0. Both TP-GMM-DS and TP-proMP have large endpoints distances. TP-GPR-DS starts its movement in the opposite direction.

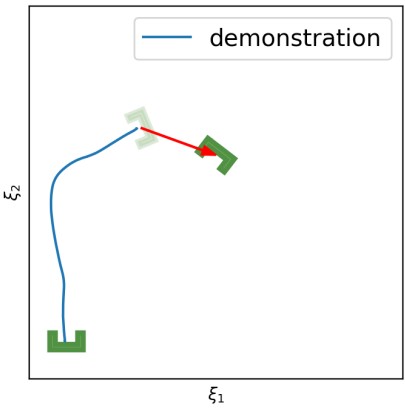

Figure 17: The single demonstration (in blue) goes from the bottom to the top, constrained by the two geometric descriptors. In the new scenario, the goal/exit geometric descriptor is shifted to a closed position, as indicated by the red arrow. The start/entry geometric descriptor remains at the same pose. We will need to generate a new motion policy that adapts to such a change.

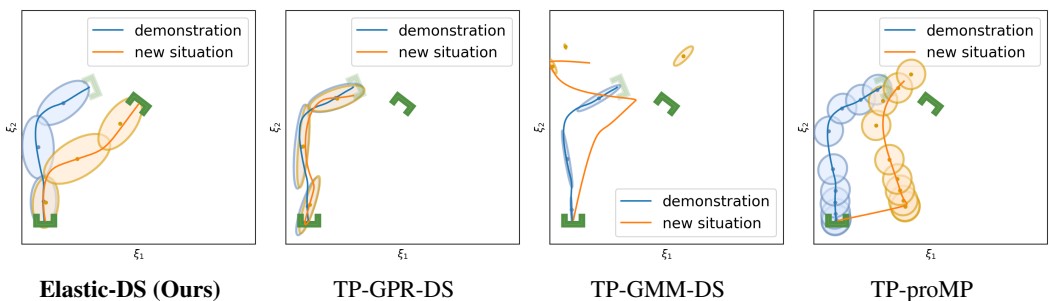

**Elastic-DS (Ours)**     TP-GPR-DS     TP-GMM-DS     TP-proMP

Figure 18: The demonstration and its probabilistic encoding are indicated as blue. The new motion policy and its probabilistic encoding for the new situation are in orange. While the other benchmark methods fail to meet the constraints given only one demonstration, Elastic-DS can generalize to the geometric descriptors constraints on both ends

## F.2 Far

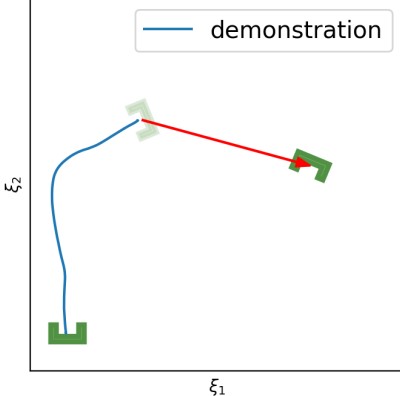

Figure 19: In the new scenario, the goal/exit geometric descriptor is shifted to a further position with rotation, as indicated by the red arrow. The start/entry geometric descriptor remains at the same pose.

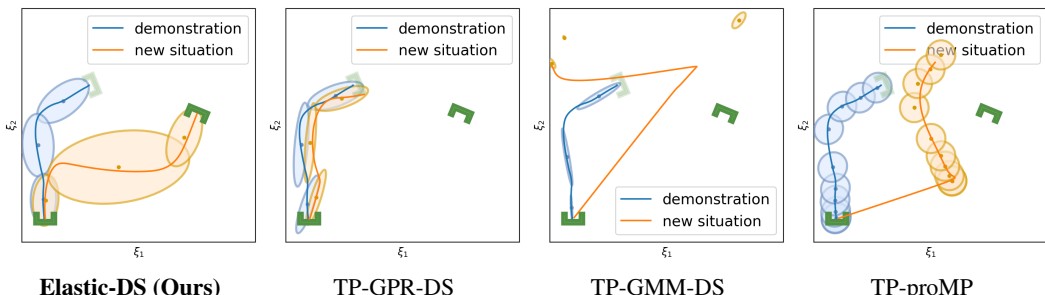

|  | **Elastic-DS (Ours)** | TP-GPR-DS | TP-GMM-DS | TP-proMP |
|:---:|:---:|:---:|:---:|:---:|

**Elastic-DS (Ours)**     TP-GPR-DS     TP-GMM-DS     TP-proMP

Figure 20: Only Elastic-DS generates a new motion policy that meets the new geometric descriptor constraints

| Metric | **Elastic-DS (Ours)** | TP-GPR-DS | TP-GMM-DS | TP-proMP |
|:---:|:---:|:---:|:---:|:---:|
| Start Cosine Similarity ↑ | **0.9971** | -0.9999 | 0.7872 | 0.3141 |
| Goal Cosine Similarity ↑ | **0.9997** | 0.5451 | 0.6724 | 0.9611 |
| Endpoints Distance ↓ | **0.0009** | 0.8459 | 1.552 | 0.6480 |

Table 3: As the goal geometric descriptor is moved further away, the performances of the other methods start to decay. The Start Cosine Similarities for TP-GMR-DS and TP-proMP decrease. Elastic-DS remains in good performance on the three metrics.

## F.3 Both Ends Shifted

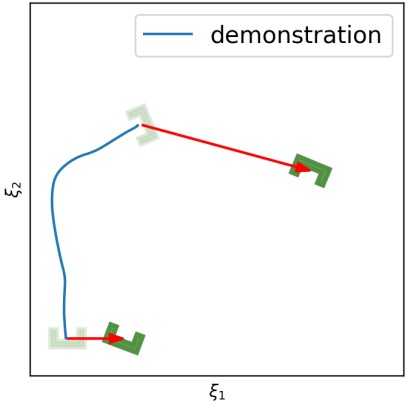

Figure 21: The new situation includes translation and rotation for both the starting and goal geometric descriptors, as indicated by the red arrows.

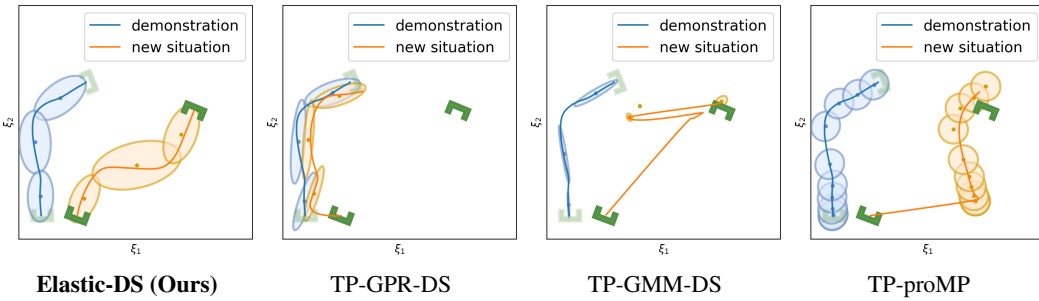

**Elastic-DS (Ours)**     TP-GPR-DS     TP-GMM-DS     TP-proMP

Figure 22: Elastic-DS is able to handle the new situation while the other benchmark methods fail.

| Metric | Elastic-DS (Ours) | TP-GPR-DS | TP-GMM-DS | TP-proMP |
|---|---|---|---|---|
| Start Cosine Similarity ↑ | **0.9843** | -0.3981 | 0.9405 | 0.4777 |
| Goal Cosine Similarity ↑ | **0.9998** | 0.5453 | 0.6324 | 0.8965 |
| Endpoints Distance ↓ | **0.0008** | 0.8360 | 0.07642 | 0.1788 |

Table 4: With both geometric descriptors moving, Elastic-DS remains in good performance on all three metrics.

## F.4 Both Ends Shifted Far

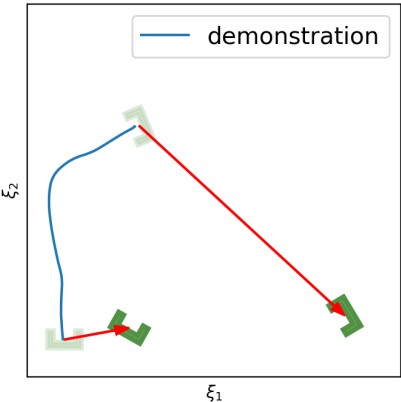

Figure 23: In the new scenario, the goal/exit geometric descriptor is shifted to an even further position with rotation, as indicated by the red arrow

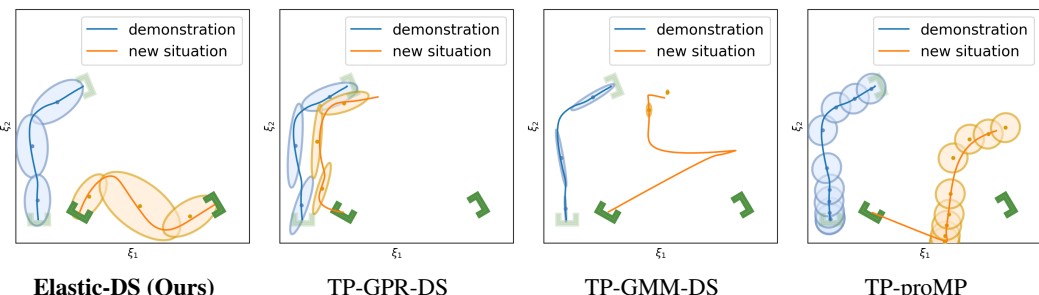

| Elastic-DS (Ours) | TP-GPR-DS | TP-GMM-DS | TP-proMP |

Figure 24: Only Elastic-DS generates a new motion policy that meets the new geometric descriptor constraints

| Metric | Elastic-DS (Ours) | TP-GPR-DS | TP-GMM-DS | TP-proMP |
|---|---|---|---|---|
| Start Cosine Similarity ↑ | **0.9869** | -0.3827 | 0.8540 | 0.1295 |
| Goal Cosine Similarity ↑ | **0.9997** | 0.9378 | 0.7151 | 0.9766 |
| Endpoints Distance ↓ | **0.0012** | 1.265 | 1.143 | 0.6468 |

Table 5: With the more challenging new scenario, the performances for the benchmark methods decay even more. The endpoint distances increase for all the TP approaches. Elastic-DS remains in good performance on all three metrics.

## G  Robot Experiments Details

### G.1  Software and Hardware Details

For all of these experiments, we used the 7DOF Franka Emika Panda robotic arm controlled via ROS and the libfranka C++ interface. The computer for the experiments ran on Ubuntu 20.04 with Intel i7-11700K 3.6GHz CPU and 32GB memory. To track the geometric descriptors $\mathcal{O}_b$ for each task, we use the Optitrack Motion Capture system, which provides us 6DoF frames of the rigid bodies at 100Hz. We first attached a set of motion capture markers to the base of the Franka Panda robot arm, which served as the fixed frame. For each experiment, we attached motion capture markers to the task-relevant objects.

To record demonstrations, we published the robot end-effector position to a ROSbag recording. During the demonstration, the orientation $R$ and position $p$ of the task-relevant objects will be recorded with the Motion Capture system. We used the finite difference of the collected position data to calculate the trajectory velocity, which forms the training data $\mathcal{D} := \{\xi_{t,n}, \dot{\xi}_{t,n}\}_{t=1}^{T_n}$.

In the training phase, we first use Elastic-GMM implemented in both MATLAB and Python to learn a GMM encoding of the training data. Then, during the testing phase, we use Elastic-GMM implemented in Python to modify the encoded data with the geometric descriptors $\mathcal{O}_i$. The Elastic-GMM output will then become the input to the MATLAB code for learning the motion policy. The execution of the Elastic-DS motion policy is implemented in a C++ ROS node, which takes the current state of the end-effector of the robot $\xi \in \mathbb{R}^3$ as the input. The output of this ROS node is the desired end-effector velocity $\dot{\xi} \in \mathbb{R}^3$, which is sent to a low-level cartesian velocity impedance controller implemented in C++ and running at 1kHz. To achieve the required velocity at the end-effector, it performs torque control at the joints. The stiffness parameter of the controller was set to 180.0. The orientation of the end-effector is fixed for every task as the Elastic-DS motion policy $\dot{\xi} = f(\xi)$ is only learned in 3D space.

### G.2  Bookshelf Experiment

The goal of this experiment is to teach the robot how to insert a book into a desktop bookshelf. With the bookshelf being moved to different locations and orientations on the table, the robot should be able to generalize and reproduce new motion policies for inserting the book into the bookshelf.

Prepare for the experiment:

1. We attached a motion capture marker object to the side of the bookshelf. The goal geometric descriptor $\mathcal{O}_g$ was at an offset from the marker object so that it was inside one of the slots in the bookshelf. The orientation $R \in SO(3)$ of the geometric descriptor $\mathcal{O}_g$ was the same as the opening of the bookshelf. Both the position $p \in \mathbb{R}^3$ and the orientation $R$ were with respect to the fixed frame.

2. Closed the gripper to hold the center of the book vertically.

3. Calibrated the weight of the book so that the robot would not move in gravity compensation.

Collect data:

1. Recorded the robot end-effector position $\xi_t \in \mathbb{R}^3$ to a ROSbag.

2. At the same moment, the motion capture system recorded the bookshelf (geometric descriptor) position and orientation $\mathcal{O}_g$.

3. As shown in the figure below, a person used hands directly in touch with the robot to perform a kinesthetic teaching demonstration. The ROSbag recording ended as the end effector position $\xi$ reached the bookshelf slot $p$, which is the position in $\mathcal{O}_g$.

4. The single demonstration (end effector position $\xi$ and time-derivative computed numerically with timestamp data $\dot{\xi}$), was then used as the training data $\mathcal{D} := \{\{\xi_{t,n}, \dot{\xi}_{t,n}\}_{t=1}^{T_n}\}$.

The training data $\mathcal{D}$ was encoded as Elastic-GMM $\{\pi_k, \mu'_k, \Sigma'_k\}_{k=1}^K$, as described in Section 4.1. There was no required tuning parameter.

Execution:

1. We moved the robot end-effector (with the book) and the bookshelf $\mathcal{O}_g$ to different configurations, as shown in the different figures below and the video. The end-effector orientation with the book was always aligned with the bookshelf opening so that a translational movement could insert the book into the bookshelf.

2. The motion capture system recorded the new configuration of the bookshelf.

3. For the updated bookshelf configuration $\mathcal{O}_{g,new}$, we updated the Elastic-GMM $\{\pi_k, \mu'_k, \Sigma'_k\}_{k=1}^K$ to the new situation and learned the Elastic-DS as described in Section 4.1.

4. The robot then executed the DS motion policy $\dot{\xi} = f(\xi)$ in the task space with a velocity-based impedance controller [58].

5. The gripper released the book once it reached the attractor $\xi_{curr} = p$.

6. Repeated with different configurations.

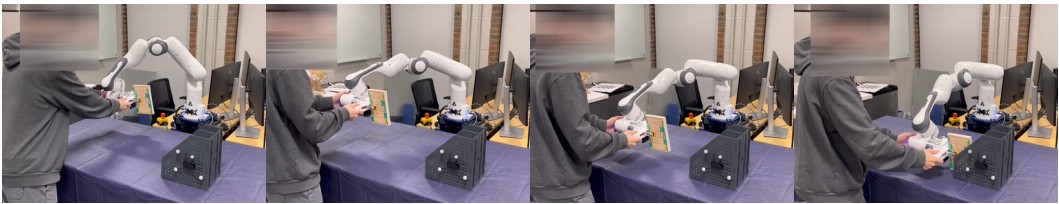

Figure 25: Demonstration for inserting book into a desktop bookshelf

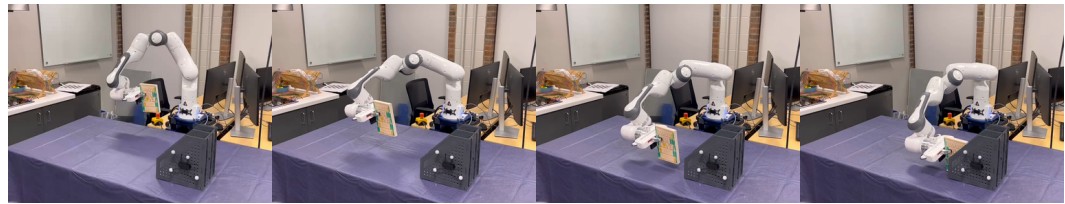

Figure 26: The execution for the learned DS in the original configuration

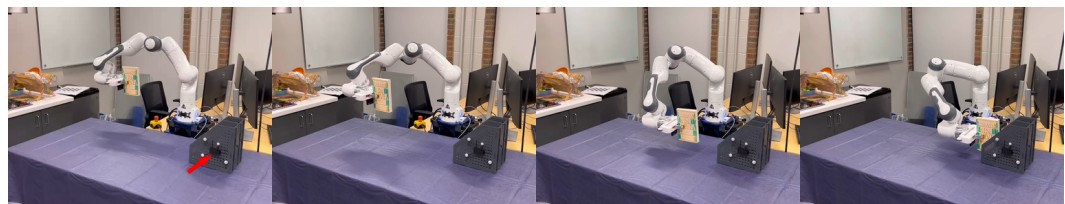

Figure 27: The bookshelf was shifted closer to the robot (The shifting direction is indicated by the red arrow). Without any new demonstration, the learned DS was able to adapt to the new configuration.

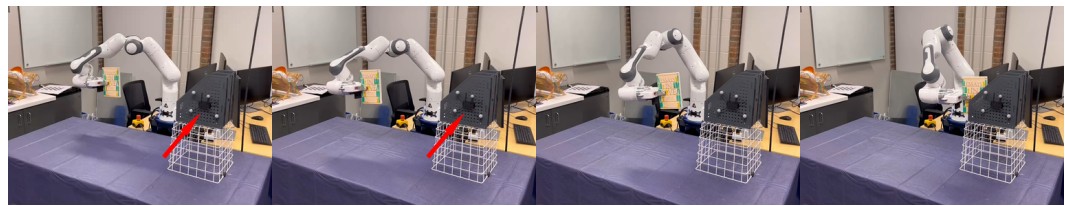

Figure 28: The bookshelf was shifted up (The shifting direction is indicated by the red arrow). Without any new demonstration, the Elastic-DS was able to adapt to the new configuration.

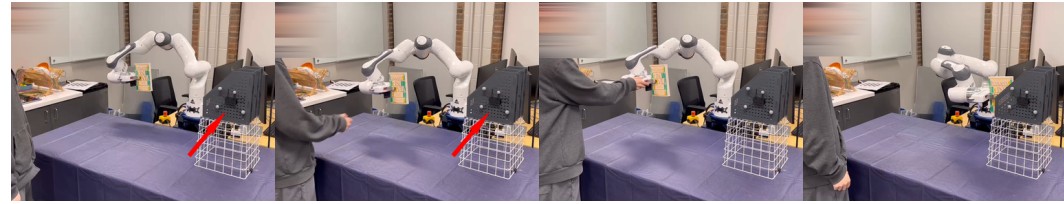

Figure 29: The bookshelf was shifted up (The shifting direction is indicated by the red arrow). Without any new demonstration, the Elastic-DS was able to adapt to the new configuration. The robot was still able to reach the new bookshelf position with human disturbances

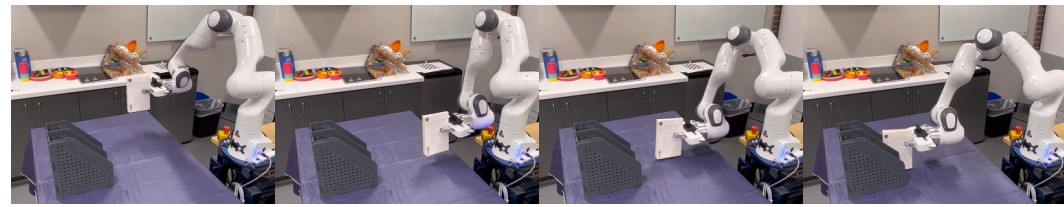

Figure 30: The bookshelf is rotated and shifted to the left side of the robot. Without any new demonstration, the Elastic-DS was able to adapt to the new configuration. The robot was still able to reach the new bookshelf configuration. We rotate the end-effector to be parallel with the bookshelf beforehand.

### G.3 Pick and Place Experiment

In this task, we will show the robot how to pick and place a cube in a bin. The cube position $p \in \mathbb{R}^3$ can be changed (labeled by the motion capture marker on the box as the geometric descriptors $\mathcal{O}_b$ while the bin position is fixed). Based on the nature of this task, we manually set two motion segments. However, the cutoff location of the two motion segments is determined automatically by the motion capture data. The first segment is the picking motion with a geometric descriptor $\mathcal{O}_1$ at the end of the trajectory (at the cube). The second segment is the placing motion with a geometric descriptor $\mathcal{O}_2$ at the beginning to ensure the robot with the cube will move upward first to reach enough height to approach the bin from the top. So there are two geometric descriptors $\mathcal{O}_1$ and $\mathcal{O}_2$ at the cube to serve as a via-point. During the demonstration, the gripper open/close is done through voice commands (with the microphone at the bottom right of the snapshots). The gripper state is memorized and associated with each segment. At the end of each segment (reaching the attractor), the robot will open/close the gripper depending on commands during the demonstration.

Prepare for the experiment:

1. We attached a motion capture marker set to a base box for placing the cube.

Collect data:

1. Record robot end-effector position $\xi_t \in \mathbb{R}^3$ to a ROSbag.

2. At the same moment, the motion capture system recorded the box (geometric descriptors ) position $p_1$ and $p_2$ from $\mathcal{O}_1$ and $\mathcal{O}_2$. This became the cutoff of the two motion segments. As shown in the figure below, a person used hands directly in touch with the robot to perform a kinesthetic teaching pick and place demonstration in a single trajectory.

3. The human used voice command (with the mic at the bottom of the figures) to control the gripper state (open/close). The ROSbag recorded the gripper state.

4. The single demonstration (end-effector position, timestamp data gripper state) was then separated into two parts $\mathcal{D} := \{\{\xi_{t,n}, \dot{\xi}_{t,n}\}_{t=1}^{T_n}\}_{n=1}^{N=2}$ and used for training. The two segments of training data were encoded as two Elastic-GMM $\{\{\pi_{k,i}, \mu'_{k,i}, \Sigma'_{k,i}\}_{k=1}^{K}\}_{i=1}^{N=2}$ as described in Section 4.1. There was no required tuning parameter.

Execution:

1. We moved the robot end-effector and the box to different positions, as shown in the different figures below. The gripper always pointed downward at all time.

2. The motion capture system recorded the new positions of the box. It served as the new geometric descriptors' positions $p_1$ and $p_2$ from $\mathcal{O}_1$ and $\mathcal{O}_2$ as well as the switch position of the two segments. The first geometric descriptor orientation $R_1 \in SO(3)$ was always pointing down to make the gripper approach the cube from the top. The second geometric descriptor orientation $R_2 \in SO(3)$ was always pointing up to allow the gripper to reach enough height before placing the cube in the bin.

3. For the updated box (geometric descriptors) configurations $\mathcal{O}_1$ and $\mathcal{O}_2$, we updated the Elastic-GMMs $\{\{\pi_{k,i}, \mu'_{k,i}, \Sigma'_{k,i}\}_{k=1}^{K}\}_{i=1}^{N=2}$ to the new situation and learned the Elastic-DSs for the two segments as described in Section 4.1.

4. The multiple DS motion policies $\dot{\xi} = \delta(f_n(\xi))$ with one-hot activation were executed in order separated by a via-point at the cube, switching of them automatically happens at the via-point as described in Appendix C.1.

5. The gripper released the cube once it reached the attractor in $f_2(\xi)$.

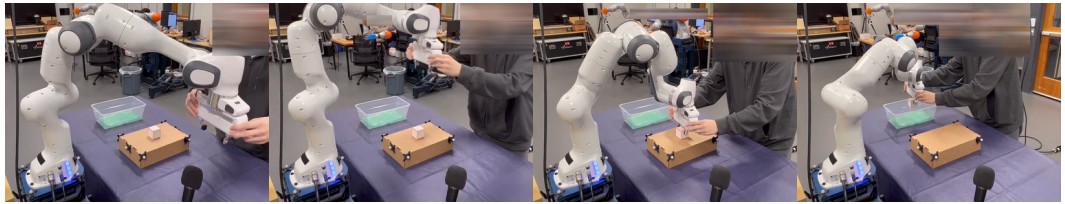

Figure 31: Demonstration for a pick and place task

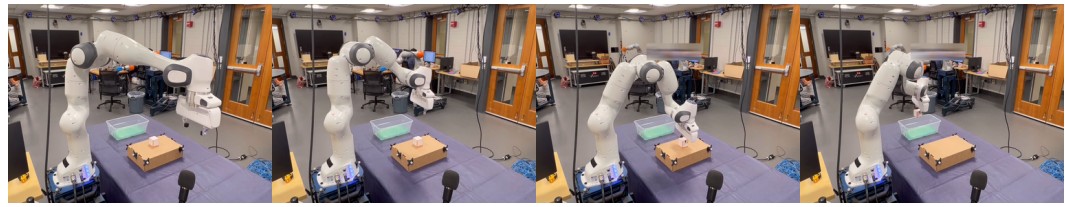

Figure 32: The execution for the learned DS in the original configuration

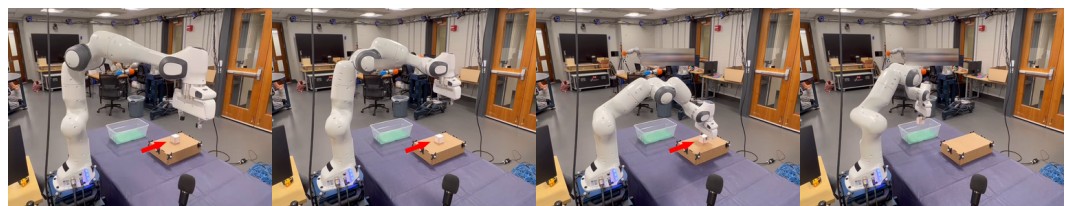

Figure 33: The cube was shifted further away from the robot (The shifting direction is indicated by the red arrow).

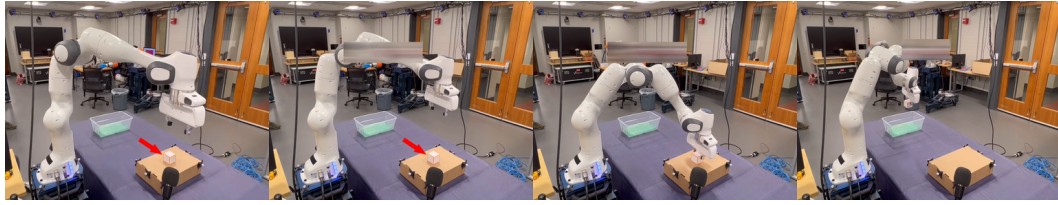

Figure 34: The cube was shifted to the right side of the robot (The shifting direction is indicated by the red arrow).

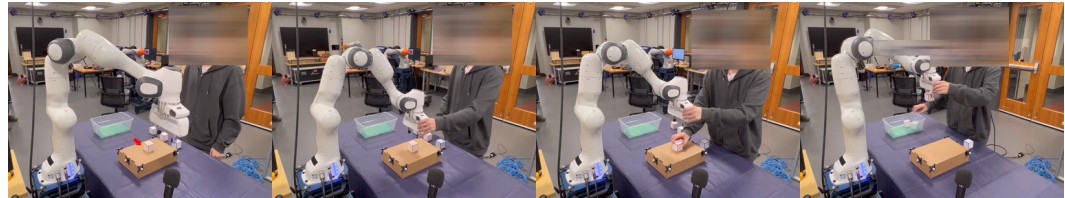

Figure 35: The cube was shifted slightly to the right side of the robot (The shifting direction is indicated by the red arrow). During the execution, the human held the robot and switched to another cube. After that, the robot finished the task

## G.4 Tunnel Experiment

In this experiment, we will show the robot how to pass through a tunnel, mimicking a scanning/inspection task. Two different motion capture marker objects label the entrance and the exit of the tunnel. Separating by the two markers, there are a total of three segments in this task. It is a task with two via points.

Prepare for the experiment:

1. We attached two motion capture objects to two sides of the tunnel (Each object has three markers).

Collect data:

1. Recorded robot end-effector position $\xi_t \in \mathbb{R}^3$ to a ROSbag.

2. At the same moment, the motion capture system recorded the entry and exit positions $\{p_b\}_{b=1}^{B=4}$ from $\{\mathcal{O}_b\}_{b=1}^{B=4}$. They became the two cutoffs of the three motion segments. A person then used hands directly in touch with the robot to perform a kinesthetic teaching tunnel demonstration in a single trajectory.

3. The single demonstration (end-effector position and timestamp data) was separated into three segments $\mathcal{D} := \{\{\xi_{t,n}, \dot{\xi}_{t,n}\}_{t=1}^{T_n}\}_{n=1}^{N=3}$ for individual training. The three segments of training data were encoded as three Elastic-GMMs $\{\{\pi_{k,i}, \mu'_{k,i}, \Sigma'_{k,i}\}_{k=1}^{K}\}_{i=1}^{N=3}$ as described in Section 4.1. There was no required tuning parameter.

Execution:

1. We moved the robot end-effector and changed the tunnel position and orientation. The gripper always pointed downward.

2. The motion capture system recorded the new positions $\{p_b\}_{b=1}^{B=4}$ of the tunnel entry and exit. They served as the new geometric descriptor position in $\{\mathcal{O}_b\}_{b=1}^{B=4}$ as well as the switch position of the three segments. The first segment had a geometric descriptor $\mathcal{O}_1$ at the end (at the tunnel entry). The second segment was within the tunnel, so it had two geometric descriptors $\mathcal{O}_2$ and $\mathcal{O}_3$ at two ends. The third segment had a geometric descriptor $\mathcal{O}_4$ at the beginning (at the tunnel exit). All of the geometric descriptors $\{\mathcal{O}_b\}_{b=1}^{B=4}$ were predefined to point along the tunnel movement direction. They will change based on the relative position of the entry and exit.

3. We updated the three Elastic-GMMs $\{\{\pi_{k,i}, \mu'_{k,i}, \Sigma'_{k,i}\}_{k=1}^{K}\}_{i=1}^{N=3}$ to the new tunnel configurations and learned a **single** Elastic-DS $\dot{\xi} = f(\xi)$ as in Appendix C.2. Except for the flip tunnel case, where we combined $\{\pi_k, \mu'_k, \Sigma'_k\}_{k=1}^{K}\}$ to learn a sequential Elastic-DS $\dot{\xi} = \delta f_n(\xi)$ as in Appendix C.1.

4. Execution of the motion policy $\dot{\xi} = \delta f_n(\xi)$ via the cartesian velocity impedance controller

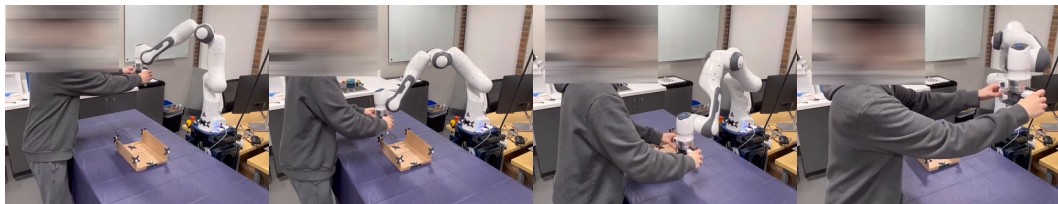

Figure 36: The human guides the end-effector for an inspection task, starting from the left side, passing through a tunnel, and stopping on the right side

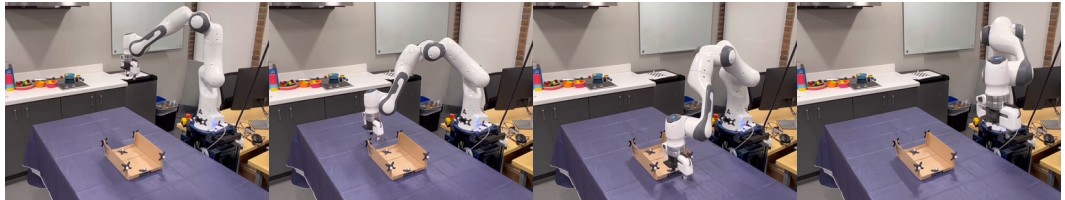

Figure 37: The execution for the learned DS in the original configuration from the demonstration

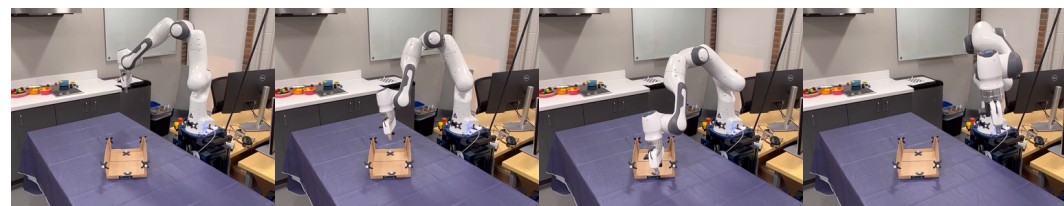

Figure 38: The tunnel is rotated. We rotate the end-effector to be parallel with the tunnel before the execution.

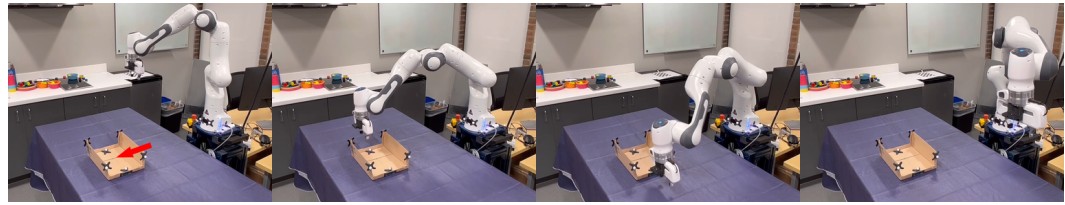

Figure 39: The tunnel is shifted further away from the robot, indicated by the red arrow.

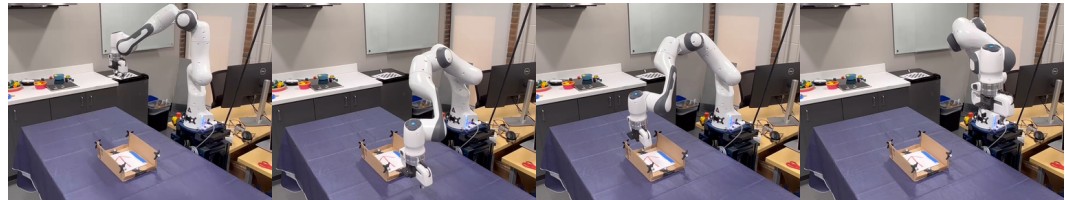

Figure 40: The tunnel is flipped, indicated by the arrow in the tunnel (as opposed to the direction from the demonstration). The robot needs to move to the right side to enter the tunnel and exit to the left side before reaching the end pose.

### G.5 Combined Experiment (Tunnel + Pick and Place)

There is no demonstration or training in this task. We reuse the Elastic-GMMs (each as $\{\{\pi_{k,i}, \mu'_{k,i}, \Sigma'_{k,i}\}_{k=1}^K\}_{i=1}^N$) learned from the previous experiments to compose new sequences which perform new tasks. We manually defined the sequence of the task with the one-hot encoding activation $\delta(\xi, o_i)$. However, in the future, we plan to develop high-level planning algorithms to determine the sequence automatically.

Prepare for the experiment:

1. We used all the previous components except for the bookshelf. The motion capture markers were placed in the same way as in the previous experiments. This time, we also added markers to the bin for the cube placing. There were a total of 7 geometric descriptors $\{\mathcal{O}_b\}_{b=1}^{B=7}$

2. We defined the sequence of the execution (Pick-Scanning-Place) in $\delta(\xi, o_i)$. There were a total of four motion segments in this task, corresponds to $\{f_n(\xi)\}_{n=1}^{N=4}$.

Execution:

1. We moved the robot end-effector and changed the object positions. The gripper always pointed downward.

2. The motion capture system recorded the new positions of the objects. The geometric descriptors $\{\mathcal{O}_b\}_{b=1}^{B=7}$ were updated.

3. We updated the four Elastic-GMMs $\{\{\pi_{k,i}, \mu'_{k,i}, \Sigma'_{k,i}\}_{k=1}^K\}_{i=1}^{N=4}$ to the new geometric descriptors' configurations $\{\mathcal{O}_b\}_{b=1}^{B=7}$ based on the motion capture data and learned four Elastic-DSs $\{f_n(\xi)\}_{n=1}^{N=4}$, as described in Section 4.1.

4. Execution of the Sequential Elastic-DS motion policy $\dot{\xi} = \delta f_n(\xi)$ (Appendix C.1) via the cartesian velocity impedance controller

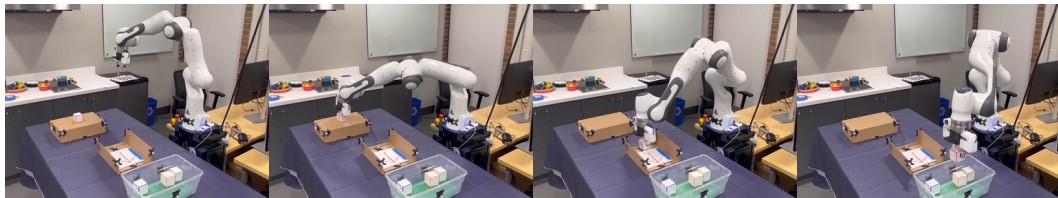

Figure 41: Composing the learned DSs with task transfer parameters from the "pick and place" and "tunnel" tasks. The robot is able to pick up a block, pass through the tunnel for scanning, and place the block in the bin. The entire motion does not require extra demonstration. Note the positions of the objects are not the same as in the original demonstrations.

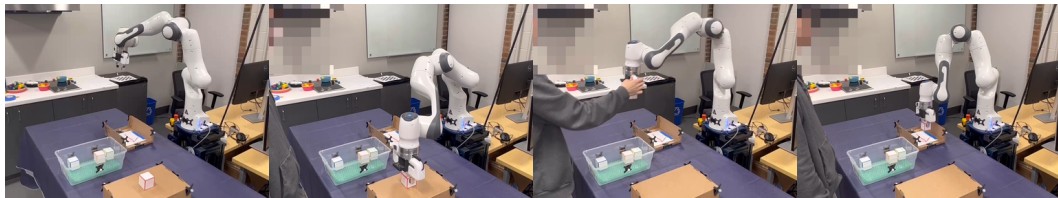

Figure 42: We shift the cube starting platform, the tunnel, and the bin. By reusing and composing the previously learned DS with task transfer, the robot is able to finish the tasks of picking, scanning, and placing without new demonstration in this new environment configuration. Also, there is human disturbance involved during the task execution.

## G.6  TP-GMM + LPV-DS vs. Elastic-DS

The inputs for LPV-DS include a GMM $\{\pi_k, \mu'_k, \Sigma'_k\}_{k=1}^K$ and a velocity reference $\dot{\xi}_t$, which the TP-GMM approach is able to provide. Therefore, one could imagine using TP-GMM with LPV-DS to create generalizable motion policies. This section showcases a robot experiment in the bookshelf case using TP-GMM + LPV-DS (TP-LPV-DS). We reused the single demonstration $\mathcal{D}$ from the bookshelf case in the previous section. Then, the bookshelf was moved and rotated to a new instance different from the previous section. After encoding the demonstration in TP-GMM and calculating the product GMM at the new instance $\mathcal{O}_b$, we used GMR to produce the velocity reference and then generated a motion policy with LPV-DS. With the single demonstration $\mathcal{D}$, TP-LPV-DS was not able to complete the book insertion task as shown in Figure 43. Therefore, more demonstrations were provided in different contexts for TP-LPV-DS to help generate a successful new policy for the new instance, as shown in Figure 44. On the other hand, using the single demonstration, Elastic-DS was able to complete the book insertion task at the new instance, as shown in Figure 46. The preparation and data collection process was the same as G.2. A single demonstration was collected.

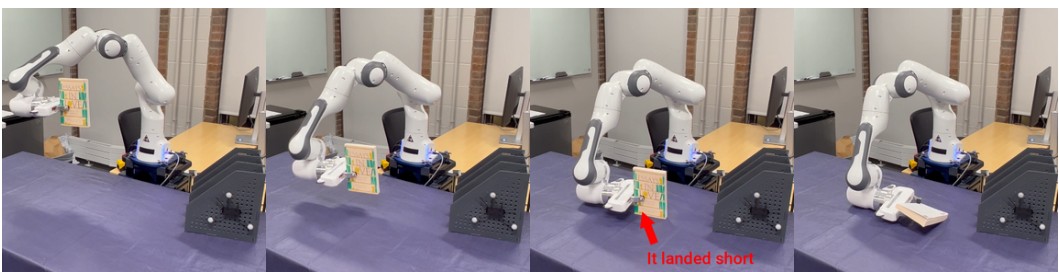

Figure 43: The bookshelf was shifted and rotated to a configuration $\{\mathcal{O}_b\}$. The execution steps were the same as G.2 except that at step 4, the new products of TP-GMM $\{\pi_k, \mu'_k, \Sigma'_k\}_{k=1}^K$ was calculated for LPV-DS. The robot then executed the DS motion policy $\dot{\xi} = f(\xi)$ in the task space with a velocity-based impedance controller [58]. After training with one demonstration, when the bookshelf was set to a new configuration, TP-LPV-DS landed short and hit the table.

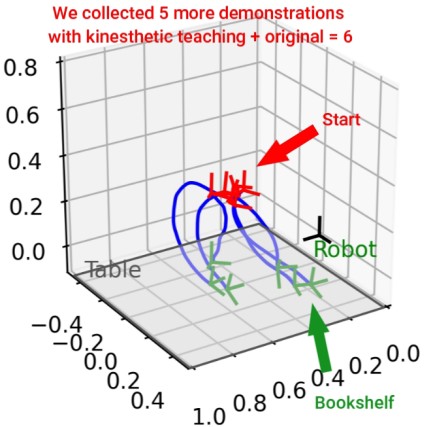

Figure 44: From the original single demonstration bookshelf configuration, we rotated the bookshelf to 2 more different configurations and collected new demonstrations. However, these were not sufficient for generating the successful motion policy for $\mathcal{O}_b$. We then collected 3 more demonstrations with different end poses closer to $\mathcal{O}_b$. The figure shows all the new demonstrations trajectory and all endpoints geometric descriptors $\{\mathcal{O}_{sj}, \mathcal{O}_{gj}\}_{j=1}^{J=6}$.

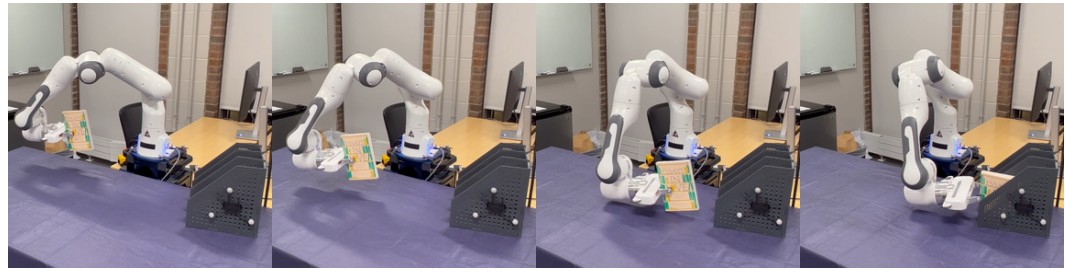

Figure 45: After training with multiple demonstrations, it used the same execution steps as the single demonstration case. When the bookshelf was set to a new configuration, TP-LPV-DS successfully completed the task.

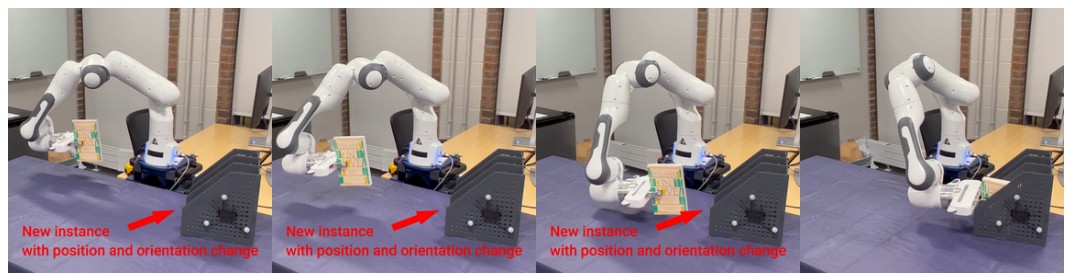

Figure 46: After training with a single demonstration, The execution steps were identical to G.2 with bookshelf configuration $\mathcal{O}_b$. Elastic-DS successfully completed the task.

# H   Training and Adaptation Computation Times

Training and adaptation of the Elastic-DS are performed on a laptop with Intel i7-12700H and 16GB memory. Initial training time for a single demonstration with roughly $T_n = 200$ datapoints is:

- Original PC-GMM implementation in Matlab [22] takes around 2-4 seconds.
- An improved parallelized PC-GMM implementation in C++ takes around 100-200ms.

For parameter adaptation, the recorded computation times are below (considering 3-4 Gaussians):

- Elastic-GMM parameter transfer takes around 30ms-80ms in Python.
- DS parameter learning (SDP optimization) takes around $\approx 800ms$ in Matlab.

Hence, for a single demonstrations with $T_n = 200$ datapoints initial training is $< 1s$ whereas generating a new policy from task parameter changes takes around 1-2s. The robot experiments presented in this section contain between $T_n = [500, 1000]$ with $\xi \in \mathbb{R}^3$. For such datasets the average computation time to generate a new policy is $\approx 4s$. In Fig. 47 we plot the trend of such computation times as function of increasing $T_n$.

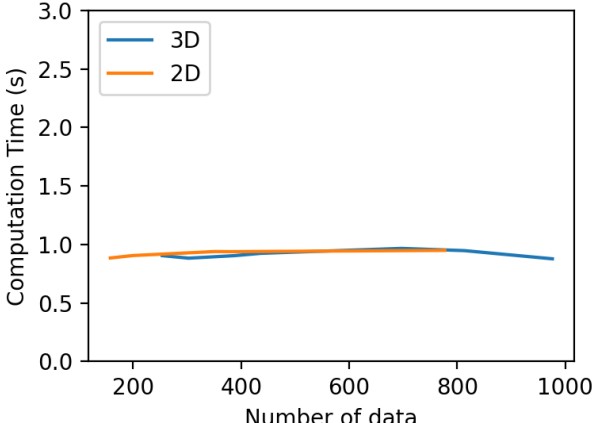

Figure 47: This plot shows the trend of Elastic-DS computation time for generating a new policy under different lengths of demonstration $T_n$ in 2D and 3D. Each data point is collected from an average of 5 runs.

