# OpenReview forum: "Task Generalization with Stability Guarantees via Elastic Dynamical System Motion Policies"
_robot-learning.org/CoRL/2023/Conference — CoRL 2023 Poster_

### Official Review · Reviewer_uBxH · 2023-07-03

**Confidence:** 4
**Originality:** Good
**Technical Quality:** Very Good
**Clarity Of Presentation:** Very Good
**Impact:** 3

**Recommendation:**

Strong Accept: I recommend accepting the paper and will argue for my recommendation even if other reviewers hold a different opinion.

**Review:**

In general, the ideas and methods presented in the paper are interesting and aim at solving the relevant problem of generalization when learning robot skills from demonstrations. The combination between DS --- which are learned from few demonstrations while offering stability guarantees --- with generalization capabilities is generally elegant. I also appreciate that the proposed approach is validated both in 2D examples and with various real-world robotic tasks.

The idea of elastic GMMs is generally elegant and seems efficient to generalize GMMs to new start- and end-points. However, I missed a clear connection between the Laplacian editing primer (Section 4.2.2) and the transformations of Gaussians described in Section 4.1.3 (beyond the fact that both involve waypoints and a least square formulation). More precisely, how are the graph Laplacian matrix $L$ and the Laplace coordinate matrix $\Delta$ computed? How do they relate to the Laplacian editing primer?

Moreover, the solution to Eq. (7) produces new joint positions $\beta_i$. However, it doesn't seem to produce new covariances $\Sigma_t$ although these covariances seem to control the orientation of the next Gaussian component in the chain. Instead, the covariances seem to simply be scaled as a function of the distance between neighboring joints. Therefore, how is the orientation of the GMM component modified?

Although the motion policy proposed in Eq. (2) describes multi-step sequential task, this aspect is only briefly discussed (Section 4.3) and experimented (in Appendices C, D.1.3) later on. The paper assumes that the sequence in created manually by appending several tasks together. In this case, how are the activation functions designed to cope with the fact that the velocity is 0 at the end of each DS? How depend is the final execution of the design of these activation functions? Which system (Fig. 5b and 5c) is used in Appendix D.1.3 and in the video?

Overall, the experimental results are very briefly shown in the paper and the reader must refer to Appendix to get an overall idea of the performance of the proposed approach. Given that the fact that the proposed DS can deal with via-points constraints and with sequences is given a high importance by being part of the general motion policy given by Eq. 2, I would have liked to see the related results in the main paper. In general, a comparison of the generalization capabilities of the proposed approach compared to the generalization capabilities of movement-primitives-based approaches (based on via points or task parameters) would strengthen the paper. Similarly, the proposed elastic GMM presents some similarities with TP-GMM as both adapt the Gaussian components of the GMM based on task parameters. One could therefore ask how would a TP-GMM compare to the proposed elastic GMM if used within LVP-DS? Although the main difference may be the fact that TP-GMM requires demonstrations in different context, the two approaches may also display different generalization capabilities.

Although the contributions of the paper become clear in Section 2, I find the title and introduction misleading in that regard. While the paper is clearly concerned with generalization over instances of the same task, the title, abstract, and introduction suggest that the proposed approach instead aims a performing task transfer, which is generally understood as transferring knowledge between **different** tasks in the context of transfer learning. The sentences "lack the flexibility to transfer to new tasks" (line 4), "learning from single tasks" (line 37), "humans tend to borrow knowledge from prior experience of skill acquisition and reuse them for acquiring and executing new tasks" (lines 38-9), "to achieve transfer" (line 94) also imply a transfer between different tasks, which does not correspond to the focus of the paper. I would suggest to rephrase the title, abstract, and introduction with a focus on generalization instead of on task transfer / transfer to novel tasks to better reflect the frame and contributions of the paper.
Along a similar line, the sentence "the robot should intelligently understand the objective, which means [...]" (line 40) is not reflected in the paper, as the important task parameters are provided by hand.


Minor:
- Line 27-8: "LfD, also known as imitation learning": LfD is a type of imitation learning instead of being a synonym of it.
- End of line 63: while ii) requiring -> iii) instead
- Missing spaces after some section title (e.g., section 2), and above/below equations (e.g., Eq. (1) )


------------
Post-rebuttal comments:

The rebuttal answered my initial concerns about the paper. Several theoretical aspects were clarified, and I believe that the contributions and scope of the paper --- in relation to generalizations over instances of the same task --- are properly reflected in the title, abstract, and introduction. The sequential aspect of the approach is now better covered in the main text. Finally, the added comparisons with state-of-the-art successfully demonstrate the advantages of the proposed approach.

Following the comment of Reviewer 4LLc, extensions of the proposed approach to orientation data would be valuable as future work.



**Quality Of The Limitations Section:**

Limitations are addressed clearly

**Questions For Rebuttal:**

- Clarifications concerning the Gaussian transformations presented in Section 4.1.3;
- Clarifications concerning the sequential aspect of the presented approach;
- Extension of the results presented in the main paper;
- Potential comparisons with respect to the generalization capabilities of state-of-the-art LfD approaches;
- Clarify the focus and contributions of the paper in the title, abstract, and introduction.

**Robotics Focus:**

Sufficient demonstration on hardware

**Summary Of Paper:**

This paper proposes to endow the linear parameter varying dynamical system (LVP-DS) formulation with generalization capabilities. To do so, it presents an elastic Gaussian mixture model (GMM) to update the underlying GMM using task parameters (start-, end-, and eventually via-points). The updated GMM is then used as a basis for an updated LVP-DS that generalizes to shifted and rotated instances of the initially-demonstrated task.

**Summary Of Recommendation:**

The paper presents interesting ideas and is generally convincing, but requires several additional explanations and some polishing before being ready for publication. Therefore, I now suggest to weakly reject the paper, but would be glad to update my score if the rebuttal answers my main concerns.

------------
Post-rebuttal:

As my main concerned have been answered by the rebuttal, I updated my score from "weak reject" to "strong accept".

---

### Official Review · Reviewer_KgCv · 2023-07-21

**Confidence:** 2
**Originality:** Good
**Technical Quality:** Good
**Clarity Of Presentation:** Good
**Impact:** 2

**Recommendation:**

Weak Reject: I recommend rejecting the paper, but will not argue for my recommendation if the majority of other reviewers have a different opinion.

**Review:**

The main strengths of the paper are that while it lacks experimental detail, the idea promises to push forward research efforts in LfDs with a particular focus on generalization, stability and effort. This is addressed by demonstrating particularly a zero-shot method. Overall, the paper is well-written and is of a good quality. The main drawbacks are the lack of experimental details and lack of stability verification.

**Quality Of The Limitations Section:**

Limitations are addressed clearly

**Questions For Rebuttal:**

(1) How do you demonstrate stability guarantees in your real experiments?
(2) Could you give more detail in the appendix on your experiments s.t. your results could be more reproducible?



**Robotics Focus:**

Sufficient demonstration on hardware

**Summary Of Paper:**

The authors propose Elastic-Dynamical System (DS), a novel DS learning and transfer approach that embeds task parameters into the Gaussian Mixture Model (GMM) based Linear Parameter Varying (LPV) DS formulation. Their motivation for proposing this is to address the shortcomings of current methods in LFD i.e. generalization vs. stability vs. effort. Their method promises i) stability guarantees, ii) the flexibility to transfer skills across novel scenarios, while iii) requiring minimal human effort during learning and transfer. They demonstrate their work in 2D experiments and also with the real robot (the Franka Emika 7 DoF manipulator in the following scenarios: Bookshelf, Pick and Place, Tunnel, and Combination).

**Summary Of Recommendation:**

I think the paper will need further details before I can recommend an acceptance.

---

### Official Review · Reviewer_4LLc · 2023-07-21

**Confidence:** 4
**Originality:** Good
**Technical Quality:** Good
**Clarity Of Presentation:** Good
**Impact:** 3

**Recommendation:**

Strong Accept: I recommend accepting the paper and will argue for my recommendation even if other reviewers hold a different opinion.

**Review:**

The paper has a good introduction and related works.
The motivation is convincing. However, it can work on clarifying the significance and experimental evaluations with respect to the state-of-the-art.

Strengths:
The paper addresses a very important problem in robot learning from demonstration. The generalization of learnt policy in new situations is formulated as a dynamical system with stability and convergence guarantees.
The experiments on real robot demonstrate the validity of the approach

Weakness:
The significance of the proposed is not so well placed with respect to the state-of-the-art.
The experimental test cases are relatively simple (no orientation are considered)

**Quality Of The Limitations Section:**

Additional details required

**Questions For Rebuttal:**

The paper should compare and establish some quantitative metric for comparison against established task-parameterized motion policy learning in robotics e.g. [33], [36]. Consider also the time constraint on finding the motion in a new situation.

**Robotics Focus:**

Sufficient demonstration on hardware

**Summary Of Paper:**

The paper proposes a dynamical system (DS) based task parameterized policy using  geometric descriptors for generalization. To generate the task parameters for the corresponding spatial change in the geometric descriptors, the paper proposes an augmented LPV-DS [22] in a developed dynamical system addressed as Elastic-DS. The deformation of this DS for new situation utilizes Laplacian Trajectory Editing [47].  By showing both 2D simulation and different 3D robot experiments, the paper validates the ability of Elastic-DS to perform task generalization as well as the potential for multi-task and long-horizon motion policies.

**Summary Of Recommendation:**

The paper is interesting and proposes guaranteed policy learning. However, The significance of the proposed approach with respect to the state-of-the-art is not highlighted clearly in terms of task performance. Therefore it is difficult to judge the contribution at the current stage.

-----------
Update: I think the reviewers have done a great job during rebuttal. My concerns are well addressed. I recommend acceptance.

---

### Author Response · Authors · 2023-08-12
**Rebuttal Summary**

We thank all reviewers for their constructive feedback, which has resulted in a series of changes in the paper, so far. We summarize the key changes here:

**Comparative Analysis to State-of-the-Art**: As the reviewers suggested, we added both qualitative and quantitative comparisons to the state-of-the-art task-parameterized approaches. These new results can be found in Section 5.1 in the main text and in the new Appendix E and F in the attached revised manuscript. We show that our method outperforms these baselines in terms of i) out-of-distribution generalization, ii) sample complexity (only one demonstration needed), iii) no parameter tuning (no human supervision/model selection needed) and iv) offering Lyaponuv stability guarantees.

**Clarifications in Main Text and Appendix**: We have carefully addressed the suggestions and questions from the reviewers. Numerous updates and changes have been made to both the main text and the Appendix to improve our manuscript. The changes are highlighted in light blue in the updated pdfs attached to each rebuttal. If it is a new section or the major content in the section has been changed, the section title is highlighted in light blue. The changes we have made are outlined as the following:
- We updated the title, abstract, and introduction and minor wording in other sections to keep the contribution focused on task generalization.
- We made a stronger connection between Section 4.1.2 (Laplacian Editing Primer ) and Section 4.1.3 (Transform Gaussians with Constraints) by changing and matching the notations between the sections.
- We updated Section 4.1.3 (Transform Gaussians with Constraints) to clarify how to update the orientation of the Gaussians. Also, we added Appendix G, which includes an algorithm block for the Transform Elastic-GMM.
- We moved the details about indexing in Section 4.2 (Create Velocity Profile) to Appendix B.
- We rephrased Section 4.3 (title changed to Multiple Segments) and updated Appendix C to clarify the two approaches of stitching segments.
- For Section 5.1 (2D Experiment), We added a multiple-segment example in the main text to reflect the ability to perform the via-point modification with our approach in Figure 6.
- Also for Section 5.1 (2D Experiments), we added Figure 7 and Tabel 1 to show a qualitative and quantitative comparison with other methods. There are more comparisons in different instances provided in Appendix F. We also created Appendix E to show the failure cases of TP-GMM.
- For Section 5.2 (Robot Experiments), we improve Appendix D (Robot Experiments Details) by providing more information about the hardware, software, computation times, as well as thorough descriptions of the steps for each experiment.

**Change in the paper title**: After carefully considering the reviewers’ feedback and suggestions, we believe that a minor change in the title would better represent the contribution of our work: Task **Generalization** with Stability Guarantees via Elastic Dynamical System Motion Policies. We hope the proposed new title will provide better clarity and insights into the main focus of our work: Learning generalizable motion policies in the space of task parameters with stability guarantees and minimal data. We have also updated the abstract and introduction to reflect such changes. The program chairs have approved the change.

With these changes, we believe to have provided all that was requested by the reviewers, and looking forward to discussions and further feedback.

---

> ### Author Response · Authors · 2023-08-16
> **Project Website**
>
> We have created a website for the project to upload more robot experiments videos.
>
> https://sites.google.com/view/elastic-ds
>
> The original robot experiments video is uploaded with the new title of the paper.
>
> We added a new robot experiment (Website Update #1) that shows the target converging behavior of the newly generated motion policy for new instances using Elastic-DS. The video shows that the task can still be completed under large unexpected disturbances from a human.
>
> We also added a new comparison video between TP-GMM + LPV-DS and Elastic-DS in the real robot experiment (Website Update #2). We will provide further details and comparisons in the appendix of the final paper upon acceptance.
>
> We thank all reviewers again for their valuable time and insightful feedback.

---

### Decision · Program_Chairs · 2023-08-30

**Decision:**

Accept (Poster)

**Comment:**

This paper proposes a LfD approach that is able to adapt to new tasks without requiring new demonstrations. Overall the reviewers found that the work addresses an important problem and is a clear contribution to the field. As such, I am happy to recommend acceptance.